# Learning Language Representations with Logical Inductive Bias

**Jianshu Chen**
Tencent AI Lab, Bellevue, WA 98004, USA
`jianshuchen@global.tencent.com`

## Abstract

Transformer architectures have achieved great success in solving natural language tasks, which learn strong language representations from large-scale unlabeled texts. In this paper, we seek to go further beyond and explore a new *logical inductive bias* for better language representation learning. Logic reasoning is known as a formal methodology to reach answers from given knowledge and facts. Inspired by such a view, we develop a novel neural architecture named FOLNet (**F**irst-**O**rder **L**ogic **Net**work), to encode this new inductive bias. We construct a set of neural logic operators as learnable Horn clauses, which are further *forward-chained* into a fully differentiable neural architecture (FOLNet). Interestingly, we find that the self-attention module in transformers can be composed by two of our neural logic operators, which probably explains their strong reasoning performance. Our proposed FOLNet has the same input and output interfaces as other pretrained models and thus could be pretrained/finetuned by using similar losses. It also allows FOLNet to be used in a plug-and-play manner when replacing other pretrained models. With our logical inductive bias, the *same* set of "logic deduction skills" learned through pretraining are expected to be equally capable of solving diverse downstream tasks. For this reason, FOLNet learns language representations that have much stronger transfer capabilities. Experimental results on several language understanding tasks show that our pretrained FOLNet model outperforms the existing strong transformer-based approaches.[1]

## 1 Introduction

Pretrained transformer models have achieved great success in solving natural language tasks, which learn strong language representations from large-scale unlabeled texts. The learned representations can be easily transferred to different downstream tasks by finetuning over limited amount of labeled data (Radford et al., 2018; Devlin et al., 2018; Lan et al., 2019; Liu et al., 2019; Yang et al., 2019). They even exhibit strong zero-shot or few-shot generalization capability without finetuning when further scaling up the model size (Radford et al., 2019; Brown et al., 2020; Chowdhery et al., 2022). Besides large-scale models and training data, one important reason for the success is the strong relational inductive bias encoded in the transformer architecture (Vaswani et al., 2017); it effectively models the pairwise relations between tokens and use it to compute the language representations.

In this paper, we seek to go beyond the inductive bias in transformer models and explore a new *logical inductive bias* for better language representation learning. The main idea is to view the computation of language representations as a logic reasoning process; that is, the language representations are deduced via logic reasoning step-by-step from the original discrete token sequences. Specifically, we treat the tokens in the input sequence as the terms in logic programming, and treat their properties and relations as the predicates of different arities. Then, the final language representations are derived as the *advanced* properties and relations from the *basic* input properties and relations (e.g., token ids and relative distances). Most importantly, we require the construction of such deduction process to follow the principles of first-order logic, in order to encode such logical inductive bias.

Following the above logical inductive bias, we derive a principled neural architecture, named FOLNet (**F**irst-**O**rder **L**ogic **Net**work), for learning language representations. Specifically, we construct a set

---

[1]The code along with the pretrained model checkpoints will be released publicly.

of neural logic operators as learnable Horn clauses, which are further *forward-chained* into a fully differentiable neural architecture. In particular, the FOLNet architecture consists of two interacting branches responsible for unary and binary relational reasoning, respectively. Interestingly, we find that the self-attention mechanism can be constructed by two of our developed neural logic operators, and the entire transformer architecture can be understood as a single-branch version of FOLNet. This newly discovered connection might partially explain the surprisingly strong reasoning performance of the transformer architecture (Wei et al., 2022; Lewkowycz et al., 2022). As we will demonstrate in our experiments, such dual-branch architecture has several significant advantages that are essential for learning better language representations. Furthermore, we also establish a new unified understanding of different positional encoding strategies with our logical inductive bias. For instance, we find that the existing popular relative positional encoding can be constructed by the degenerated version of our two neural logic operators. More importantly, it also allows us to develop a new principled relative positional encoding that is simple yet quite effective in practice. Notably, our proposed FOLNet has the same input and output interfaces as other pretrained transformer models (e.g., BERT) and thus could be trained by using similar losses. It also allows FOLNet to be used in a plug-and-play manner when replacing other pretrained models in solving downstream tasks. Our logical inductive bias assumes that the "logic deduction skills" are shared across all natural language tasks; that is, these skills learned during pretraining should be equally applicable to solving diverse downstream tasks. For this reason, FOLNet learns language representations that have much stronger transfer generalization capabilities. Experimental results on several language understanding tasks (GLUE, SQuAD 2.0 and FOLIO) show that our FOLNet model outperforms the transformer architecture by a large-margin when they are pretrained using similar losses. The results clearly show that advantage of using the logical inductive bias for learning language representations.

## 2 LOGICAL INDUCTIVE BIAS FOR LANGUAGE REPRESENTATIONS

Natural language text can be viewed as a sequence of discrete symbols, and language representations learning considers the problem of mapping the discrete symbols into certain more computable forms. One widely used approach is distributed representation, which maps the discrete token ids into dense vectors (Mikolov et al., 2013; Pennington et al., 2014; Peters et al., 2018). Many different functional forms, such as LSTM (Hochreiter & Schmidhuber, 1997), and more recently, transformer models (Vaswani et al., 2017), have been used to implement such mappings. They generally encode different kinds of inductive bias for modeling natural languages. For example, RNNs use the same set of model parameters to update the hidden states over time, which encodes translation-invariance over time (Battaglia et al., 2018). These forms of inductive bias continuously push the state-of-the-arts in solving natural language tasks. In this section, we introduce a new form of inductive bias, named *logical inductive bias*, which will work together with distributed representations to design more effective representation mappings. Our main idea is to view the language representation mapping as a logic reasoning process; that is, the language representations are deduced step-by-step from the original discrete token sequences. Specifically, we treat the tokens in the input sequence as terms (or objects) in logic programming, and treat their properties and relations as predicates of different arities. In light of logical inductive bias, the language representations that we seek to compute are the (advanced) properties and relations that can be deduced from these input (basic) properties and relations. Most importantly, we require the construction of such deduction process to follow the principles of first-order logic, in order to encode the logical inductive bias into the representation learning process. We now formulate the language representation learning as a logic programming problem by adopting similar (logic programming) notations used in Evans & Grefenstette (2018).

- **Terms:** We consider a first-order logic system without function symbols, so that terms can only be variables or constant. They are used to represent general objects or a particular object of interest, respectively. In the context of language representation learning, we model each instance of text sequence $x$ (of length $T$) as a collection of constants $x = \{x_1, \ldots, x_T\}$, where each token $x_t$ is a constant ($t = 1, \ldots, T$). We use lower-case letters to denote constants and upper case for variables as in logic programming. For example, $X$ is a variable to represent a general object (e.g., token).
- **Atoms:** For each term, we will define its properties and relations as an $r$-ary predicate $p(X_1, \ldots, X_r)$, which takes the value of T (True) or F (False) depending on whether a certain property/relation regarding $(X_1, \ldots, X_r)$ holds or not. For example, whether the a token $a$ takes the $v$-th id in the vocabulary is a unary predicate $\texttt{TokenID}_v(a)$ for $v = 1, \ldots, V$, where $V$

| Language representations | Logic programming | FOLNet |
|---|---|---|
| Tokens $x_t$ in the text sequence | Constant: $x_t$ | The argument: $x_t$ in tensor $\mathbf{u}_l(x_t)$ |
| Token ids, relative distances, etc | Input (basic) atoms: $\mathcal{C}_0 = \mathcal{B}$ | Input tensors: $\{\mathbf{u}_0(x), \mathbf{u}_0(x,y)\}$ |
| Final langauge representation | Deduced (advanced) atoms: $\mathcal{C}_L$ | Output tensors: $\{\mathbf{u}_L(x), \mathbf{u}_L(x,y)\}$ |
| Representation mapping (partial) | Modus Ponens using generic clause (8) | Neural logic operator: see Table 2 |
| Representation mapping (partial) | 1-step deduction: $\mathcal{C}_l = \mathrm{con}_\mathcal{R}(\mathcal{C}_{l-1})$ | Forward pass: 1-layer |
| Representation mapping (full) | $L$-step deduction: forward-chaining (3) | Forward pass: $L$-layer |

Table 1: Identification of language representation learning as a logic programming problem.

is the vocabulary size, and whether the distance between two tokens $a$ and $b$ is equal to $d$ is a binary predicate $\mathtt{Dist}_d(a,b)$, for $|d| < T$. An atom is *ground* if it has no variables, e.g., the above $\mathtt{TokenID}_v(a)$ and $\mathtt{Dist}_d(a,b)$ are all ground atoms.

- **Clauses:** The reasoning process is constructed by a set of "if-then" clauses in the form of:
$$q \leftarrow p_1 \wedge \cdots \wedge p_m, \tag{1}$$
where $p_1, \ldots, p_m$ and $q$ are the body atoms and head atoms, respectively, and $\wedge$ denotes conjunction (i.e., logical $\mathtt{AND}$). These atoms play the roles of premises and conclusions: if $p_1, \ldots, p_m$ are true, then $q$ is also true. Clauses of the above form are known as *definite Horn clauses* (Horn, 1951). We call a clause *a ground rule* if all its variables are substituted by constants. For example, when applying a substitution $\theta \triangleq \{a/X, b/Y\}$ to a clause $q(X,Y) \leftarrow p(X,Y)$, we get a ground rule: $q(a,b) \leftarrow p(a,b)$. It can be viewed as applying a general clause to a particular instantiation.

Our objective is to learn a collection of clauses and compose them into a mapping from input predicates (e.g., $\mathtt{TokenID}_v(x_t)$ and $\mathtt{RelDist}_d(x_t, x_\tau)$) to language representations. Specifically, let $\mathcal{R}$ be a set of clauses and $\mathtt{ground}(\mathcal{R})$ be the corresponding set of ground rules. We define the *immediate consequences* of applying the ground rules in $\mathtt{ground}(\mathcal{R})$ to a set of ground atoms $\mathcal{X}$ as

$$\mathrm{con}_\mathcal{R}(\mathcal{X}) = \mathcal{X} \cup \left\{ q \middle| q \leftarrow p_1 \wedge \cdots \wedge p_m \in \mathtt{ground}(\mathcal{R}), \bigwedge_{i=1}^m p_i \in \mathcal{X} \right\}. \tag{2}$$

It can be understood as a set of ground atoms that can be deduced from $\mathcal{X}$ together with $\mathcal{X}$ itself. Given a set of input ground atoms $\mathcal{B}$, we can repeatedly apply the ground rules in $\mathcal{R}$ for $L$ steps:
$$\mathcal{C}_l = \mathrm{con}_\mathcal{R}(\mathcal{C}_{l-1}), \qquad \mathcal{C}_0 = \mathcal{B} \text{ and } l = 1, \ldots, L. \tag{3}$$
Then, $\mathcal{C}_L$ is all the possible ground atoms (predicates) that can be deduced from $\mathcal{B}$ (including $\mathcal{B}$ itself) in $L$ steps. The above procedure is known as **forward-chaining**: it deduces all the possible conclusions from the input premises (i.e., $\mathcal{B}$) by repeatedly applying (i.e., chaining) clauses. If we want to verify whether a predicate $q'$ holds (i.e., can be *entailed*), it suffices to check if $q'$ is in $\mathcal{C}_L$. In language representation learning, we start, for example, from the following input (basic) atoms:
$$\mathcal{B} = \{\mathtt{TokenID}_v(x_t), \mathtt{RelDist}_d(x_t, x_\tau), \ldots | t, \tau = 1, \ldots, T\}, \tag{4}$$
and deduce $\mathcal{C}_L$ as the final representations by forward-chaining our (learned) clauses. For example, in solving an (extractive) question answering problem, whether a certain token is the beginning (or end) of the answer span is modeled as an advanced deduced property of this token, i.e.,
$$\mathcal{C}_L = \{\mathtt{AnswerStartsAt}(x_t), \mathtt{AnswerEndsAt}(x_t) | t = 1, \ldots, T\}. \tag{5}$$
In autoregressive language modeling, the advanced deduced property becomes the next token ids. Table 1 summarizes the above identifications between language representations and logic programming. Next, we will develop a neural architecture to encode such logical inductive bias. Throughout the paper, we will use boldface letters to denote vectors (lowercase) and matrices (uppercase).

## 3 FOLNet: A Neural Architecture with Logical Inductive Bias

In this section, we develop a novel neural architecture, named **F**irst-**O**rder **L**ogic **Net**work (FOLNet), which encodes the logical inductive bias presented in Section 2. Specifically, we focus on: (i) how to represent the atoms as continuous vectors (Section 3.1), (ii) how to devise a neural inference step that approximates (2) (Section 3.2), and (iii) how to forward-chain them into a fully-differentiable architecture based on (3) (Section 3.3). The overall neural architecture and its correspondence to the logical inductive bias are shown in Figure 1. Next, we will discuss step-by-step how we devise the architecture, its advantages, and also some important connections to existing approaches.

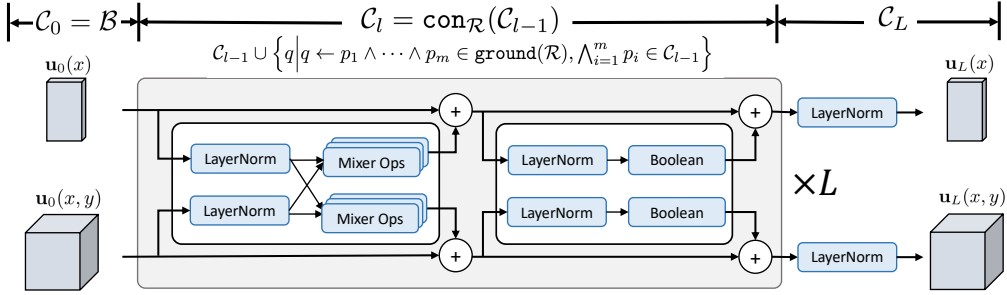

Figure 1: Overview of the FOLNet architecture and how it encodes the logical inductive bias. The neural logic operators model the clauses, which are forward-chained into a differentiable model. The "mixer ops" refer to the operators c, j, m, and t in Table 2 as they reduce over the object dimension.

## 3.1 VECTOR REPRESENTATIONS OF ATOMS

Recall that we use an $r$-ary ground atom $p_d(x_1, \ldots, x_r)$ to characterize whether the $d$-th property/relation holds for a tuple of tokens $(x_1, \ldots, x_r)$, where $d = 1, \ldots, D_r$. To overcome the difficulty of learning these discrete-valued atoms, which takes values in $\{\mathsf{T}, \mathsf{F}\}$, we introduce $u_d(x_1, \ldots, x_r) \in \mathbb{R}$ as its continuous representation and characterizes the extent to which the atom $p_d$ is true. For example, in ProbLog (De Raedt et al., 2007; De Raedt & Kimmig, 2015; Fierens et al., 2012; Manhaeve et al., 2018), $u_d(\cdot)$ gives the probability of the atom $p_d(\cdot)$ being true. In this paper, we consider $u_d(\cdot)$ to be the *logit* of the corresponding atom, i.e., $\Pr\{p_d(\cdot) = \mathsf{T}\} = 1/(1 + e^{-u_d(\cdot)})$; a larger value of $u_d(\cdot)$ implies a higher chance of the atom $p_d(\cdot)$ being true. And we can also easily verify that the logit for the negated atom of $p_d(\cdot)$ is simply $-u_d(\cdot)$. Let $\mathbf{u}(x_1, \ldots, x_r) \in \mathcal{U} \subset \mathbb{R}^{D_r}$ be a $D_r$-dimensional vector that collects $u_d(x_1, \ldots, x_r)$ as its $d$-th element. Then, $\mathbf{u}(x_1, \ldots, x_r)$ will be a continuous (logit) vector representation for $D_r$ atoms with arity $r$. For example, for an input text sequence $x = (x_1, \ldots, x_T)$, we can use $\mathbf{u}(x_t)$ to represent $D_1$ (unary) properties for each token $x_t$, and use $\mathbf{u}(x_t, x_\tau)$ to characterize $D_2$ (binary) relations between any pair of tokens. Both the input (basic) properties/relations and the deduced (advanced) ones will be represented in the same logit vector space, where the deduction process will be carried out. For convenience, we may also directly refer to a set of atoms by their logit vector representation $\mathbf{u}(\cdot)$.

## 3.2 NEURAL MODUS PONENS INFERENCE

We now develop a set of neural operators for implementing the deduction step characterized in (2). To begin with, we first introduce the *Modus Ponens* (MP) rule from first-order logic (Andrews, 2013), which states that if clause $B \leftarrow A$ and statement $A$ hold, then $B$ is true:

$$B \Leftarrow \{B \leftarrow A \text{ and } A\}. \tag{6}$$

In the context of (2), when choosing $A = p_1 \wedge \cdots \wedge p_m$ and $B = q$, the deduction in (2) can be viewed as an applications of the MP inference using all the ground clauses in $\mathtt{ground}(\mathcal{R})$. In this paper, we restrict our focus to the setting where all the atoms have arity of either one or two. That is, we will only consider atoms of the form $\mathbf{u}(x_t)$ and $\mathbf{u}(x_t, x_\tau)$ (represented in their vector forms), respectively.[2] Then, we will need to develop the MP inference from a set of atoms $\{\mathbf{v}(a), \mathbf{v}(a, b)\}$ to another set of atoms $\{\mathbf{u}(x), \mathbf{u}(x, y)\}$, which can be categorized into the following four groups:

$$\mathbf{u}(x) \Leftarrow \mathcal{R}_{\mathsf{UU}}, \mathbf{v}(a); \quad \mathbf{u}(x) \Leftarrow \mathcal{R}_{\mathsf{UB}}, \mathbf{v}(a, b); \quad \mathbf{u}(x, y) \Leftarrow \mathcal{R}_{\mathsf{BU}}, \mathbf{v}(a); \quad \mathbf{u}(x, y) \Leftarrow \mathcal{R}_{\mathsf{BB}}, \mathbf{v}(a, b) \tag{7}$$

where $\mathcal{R}_{\mathsf{UU}}$, $\mathcal{R}_{\mathsf{UB}}$, $\mathcal{R}_{\mathsf{BU}}$ and $\mathcal{R}_{\mathsf{BB}}$ denote the sets of rules that deduce atoms of a certain arity from another arity. For example, $\mathcal{R}_{\mathsf{BU}}$ defines a collection of clauses that derives binary (B) atoms from unary (U) atoms. We now proceed to model these clauses and their inference in logit space. Given a set of premise atoms $P_1, \ldots, P_M$, we consider $N$ clauses of the following generic form:

$$Q_n \leftarrow \left(\bigwedge_{m \in \mathcal{M}_{n,+}} P_m\right) \bigwedge \left(\bigwedge_{m \in \mathcal{M}_{n,-}} \neg P_m\right), \quad n = 1, \ldots, N, \tag{8}$$

---

[2]Generalizing our work to higher arities is relatively straighforward in principle but will lead to high computation complexities in practice. We leave such an extension as a future work.

| Sym. | Ops. | Typing | B-dim. | R-dim. | Neural operator in logit space | Kernel act. | Remarks |
|------|------|--------|--------|--------|-------------------------------|-------------|---------|
| b | bool | $\mathtt{U} \leftarrow \mathtt{U} \times \mathtt{U}$ | $x$ | $w$ | $\mathbf{u}_{hs}(x) = \sum_w \mathbf{K}_{hw}(x)\mathbf{v}_{ws}(x)$ | Identity | |
| c | cjoin | $\mathtt{U} \leftarrow \mathtt{U} \times \mathtt{B}$ | $h$ | $a$ | $\mathbf{u}_{hs}(x) = \sum_a \mathbf{K}_{hs}(a)\mathbf{v}_h(x,a)$ | Softmax$_a$ | |
| j | join | $\mathtt{U} \leftarrow \mathtt{B} \times \mathtt{U}$ | $h$ | $a$ | $\mathbf{u}_{hs}(x) = \sum_a \mathbf{K}_h(x,a)\mathbf{v}_{hs}(a)$ | Softmax$_a$ | Self-attention |
| m | mu | $\mathtt{U} \leftarrow \mathtt{B} \times \mathtt{B}$ | $x$ | $a$ | $\mathbf{u}_{hs}(x) = \sum_a \mathbf{K}_h(x,a)\mathbf{v}_s(x,a)$ | Softmax$_a$ | General RPE |
| a | assoc | $\mathtt{B} \leftarrow \mathtt{U} \times \mathtt{U}$ | $h$ | $w$ | $\mathbf{u}_h(x,y) = \sum_w \mathbf{K}_{hw}(x)\mathbf{v}_{hw}(y)$ | Identity | Self-attention |
| p | prod | $\mathtt{B} \leftarrow \mathtt{U} \times \mathtt{B}$ | $x$ | $w$ | $\mathbf{u}_h(x,y) = \sum_w \mathbf{K}_{hw}(x)\mathbf{v}_w(x,y)$ | Identity | General RPE |
| t | trans | $\mathtt{B} \leftarrow \mathtt{B} \times \mathtt{B}$ | $h$ | $a$ | $\mathbf{u}_h(x,y) = \sum_a \mathbf{K}_h(x,a)\mathbf{v}_h(a,y)$ | Softmax$_a$ | |

Table 2: List of all our neural logic operators with restricted kernels (see Appendix A.1 for our naming protocols). Note that each operator has a unique batching dimension (B-dim) and a unique reduction dimension (R-dim). The typing of the operator defines the arities of the kernel, the premise and the output atom. For example, $\mathtt{U} \leftarrow \mathtt{B} \times \mathtt{U}$ means the arities of the kernel, the input atom, and the output atom are 2, 1 and 1, respectively. We also list the activation functions that are used to compute the corresponding kernels, where the normalization dimension of Softmax is listed in its subscript.

where $Q_n$ is the head atom, $\neg$ denotes logical negation (i.e., NOT), and $\mathcal{M}_{n,+}$ and $\mathcal{M}_{n,-}$ are two subsets of $\mathcal{M} = \{1, \ldots, M\}$ with $\mathcal{M}_{n,+} \cap \mathcal{M}_{n,-} = \emptyset$. Then, the logit vector $\mathbf{u}$ for the head atoms $\{Q_n\}$ can be approximately inferred from the logit vector $\mathbf{v}$ of the premises $\{P_m\}$ by a matrix multiplication followed by an (optional) elementwise nonlinear activation function (Appendix D):

$$\mathbf{u} = \sigma(\mathbf{K}\mathbf{v}), \tag{9}$$

where $\mathbf{K}$ is an $N \times M$ *kernel* matrix that represents the clauses in (8), and $\sigma(\mathbf{z}) = \ln(1 + 2e^{\mathbf{z}})$ is the activation function. Notably, each row of $\mathbf{K}$ characterizes the conjunction pattern on the right-hand side of (8): a positive (negative) value of its $(n, m)$-th element, $\mathbf{K}_{nm}$, means that $m$ is more likely in $\mathcal{M}_{n,+}$ ($\mathcal{M}_{n,-}$) for the $n$-th clause. It follows that the kernels for $\mathcal{R}_{\mathtt{UU}}$, $\mathcal{R}_{\mathtt{UB}}$, $\mathcal{R}_{\mathtt{BU}}$ and $\mathcal{R}_{\mathtt{BB}}$ in (7) would be in the form of $\mathbf{K}_{\mathtt{UU}}(x,a)$, $\mathbf{K}_{\mathtt{UB}}(x,a,b)$, $\mathbf{K}_{\mathtt{BU}}(x,y,a)$ and $\mathbf{K}_{\mathtt{BB}}(x,y,a,b)$, respectively, which are $D_1 \times D_1$, $D_1 \times D_2$, $D_2 \times D_1$ and $D_2 \times D_2$ matrices. And (9) would become matrix multiplications between them and their corresponding premises (i.e., $\mathbf{v}(a)$, $\mathbf{v}(a,b)$, $\mathbf{v}(a)$ or $\mathbf{v}(a,b)$) followed by a summation over $a$ and $b$ whoever appear therein (see (30)–(33) in Appendix D). Finally, the activation function $\sigma(\cdot)$ can be dropped when "$\leftarrow$" is replaced with "$\equiv$", where $A \equiv B$ iff $A \leftarrow B$ and $B \leftarrow A$.

One major limitation of directly implementing (9) for the inference rules in (7) is the high memory and computation costs. For example, the kernel $\mathbf{K}_{\mathtt{BB}}(x,y,a,b)$ needs $O(D_2^2 T^4)$ to store its value. And the MP inference (9), which now becomes $\mathbf{u}(x,y) = \sum_{a,b} \mathbf{K}(x,y,a,b)\mathbf{v}(a,b)$, also has a computation complexity of $O(D_2^2 T^4)$. Therefore, we have to restrict the size of the kernel and reduce the overall complexity by using different methods, such as sharing the values of $\mathbf{K}_{\mathtt{BB}}(x,y,a,b)$ across different dimensions. We now provide a systematic approach based on the following principles:

1. We restrict all the kernels to be in the form of $\{\mathbf{K}(\omega), \mathbf{K}(\omega, \nu)\}$, i.e., the arity $r = 1, 2$.

2. We pick one reduction dimension and one batching dimension in the matrix multiplication.

With the above assumption, we factor the predicate dimensions of unary kernels and unary atoms so that $\mathbf{K}_d(\omega) = \mathbf{K}_{hs}(\omega)$ and $\mathbf{u}_d(x) = \mathbf{u}_{hs}(x)$, where $d = (h-1)S + s$ with $h = 1, \ldots, H$ and $s = 1, \ldots, S$. This is inspired by the multi-head attention mechanism (Vaswani et al., 2017), where $h$ is akin to the head index, $H$ is the number of heads and $S$ is the size of the head. Then, we enumerate all possible neural logic operators that can compatibly multiply a kernel from $\{\mathbf{K}(\omega), \mathbf{K}(\omega, \nu)\}$ with a premise from $\{\mathbf{v}(a), \mathbf{v}(a, b)\}$ by properly choosing different reduction and the batching dimensions. With this, we list the resulting neural logic operators for each typing in Table 2, which are further discussed in Appendix A.1 for their different roles in (restricted) Modus Ponens reasoning.

**Connection with transformers** Interestingly, we find that the j-operator and the a-operator share similar forms as the self-attention mechanism in transformers, where the a-operator computes the self-attention scores and the j-operator performs the self-attention operation. Furthermore, the m-operator and the p-operator indeed generalize the existing relative positional encodings developed in (Shaw et al., 2018), which are widely used in different transformer variants, such as in T5 models (Raffel et al., 2020). Specifically, we show in Appendix E that by making $\mathbf{v}_w(x, y)$ instance-independent, the p-operator computes the second term in equation (5) of Shaw et al. (2018), where $\mathbf{v}_w(x, y)$ play the role of $a_{ij}^K$. And by setting $\mathbf{v}_s(x, a)$ instance-independent, the m-operator computes the second

term in equation (3) of Shaw et al. (2018), where $\mathbf{v}_s(x, a)$ plays the role of $a_{ij}^V$. That is, under such *degenerated* settings, these two operators will compose the relative positional encoding developed therein. Note that their $a_{ij}^K$ and $a_{ij}^V$ are static learnable embeddings, while our $\mathbf{v}_w(x, y)$ and $\mathbf{v}_s(x, a)$ are dynamically computed for each instance (as we will discuss in Section 3.3). Therefore, our m-op and p-op can also be viewed as a more adaptive relative positional encoding, whose advantages will be further demonstrated in our experiments (Section 4).

### 3.3 FORWARD-CHAINING AND DIFFERENTIABLE LEARNING

Note that, in general, a logic operator takes a kernel $\mathbf{K}$ and a premise $\mathbf{v}$ to infer an outcome $\mathbf{u}$ (Table 2). Specifically, it "neuralizes" the logic deduction in (2) for a particular typing (e.g., $\mathtt{U} \leftarrow \mathtt{B} \times \mathtt{U}$). Applying all the neural logic operators amounts to have a full execution of (2), which is one recursion step in (3) that maps a set of $\{\mathbf{u}_{l-1}(x), \mathbf{u}_{l-1}(x, y)\}$ into $\{\mathbf{u}_l(x), \mathbf{u}_l(x, y)\}$. Therefore, we can naturally forward-chain $L$ stages of them together to create a fully-differentiable architecture that models the reasoning chain in (3). Figure 1 depicts such a forward-chaining process and also how it encodes the logical inductive bias described in Section 2. One remaining problem is how to obtain the kernels $\mathbf{K}(\cdot)$ and the premises $\mathbf{v}(\cdot)$ from our FOLNet architecture in Figure 1. We do this simply by applying two linear projections (one for the premise and one for the kernel) to the previous layer's output $\{\mathbf{u}_l(x), \mathbf{u}_l(x, y)\}$. For the kernel, we may further apply an activation function after the linear projection to compute the kernel (see Table 2 for the list of kernel activation functions for each operator). In other words, we parameterize the kernels $\mathbf{K}(\cdot)$ and the premises $\mathbf{v}(\cdot)$ by (the intermediate deduction results of) FOLNet itself. This is based on the observation that clauses are themselves predicates since $A \leftarrow B$ is defined as $A \vee \neg B$ (Andrews, 2013), where $\vee$ denotes disjunction (logical $\mathtt{OR}$). We now describe the input and the output of FOLNet in Figure 1. At the input, we can encode the discrete token ids for a token $x_t$ into vectors of the form $\mathbf{u}_0(x_t)$ by standard embedding lookup. Likewise, we also convert the (discrete) relative distance between two tokens $x_t$ and $x_\tau$ into a vector of the form $\mathbf{u}_0(x_t, x_\tau)$. The $\{\mathbf{u}_0(x_t), \mathbf{u}_0(x_t, x_\tau)\}_{t,\tau}$ will be used as vector representations of the base atoms $\mathcal{B}$ and fed into the FOLNet model (Figure 1). After $L$ layers (i.e., $L$ steps of deduction), the output $\{\mathbf{u}_L(x_t), \mathbf{u}_L(x_t, x_\tau)\}_{t,\tau}$ becomes the vector representations of $\mathcal{C}_L$ in (3), which is used as the final language representations. Therefore, our FOLNet model has the same input-output interface as other transformer models and can be used in a plug-and-play manner for solving downstream tasks. Because of this, our model can also be pretrained over large-scale texts using the same losses (e.g., MLM, NSP, SOP, etc) as other *encoder-only* models — see Appendix A.4 for how to compute these losses from $\{\mathbf{u}_L(x_t), \mathbf{u}_L(x_t, x_\tau)\}_{t,\tau}$. Our model can also be extended to the *decoder-only* and the *encoder-decoder* versions by slightly modifying the neural logic operators (see Appendix A.5), which can then be pretrained to predict the next words auto-regressively. We will conclude this section by discussing several important properties of FOLNet architecture.

**The dual-branch architecture** Note that the FOLNet model in Figure 1 has two branches: (i) a unary predicate branch for reasoning over $\mathbf{u}_l(x)$, and (ii) a binary predicate branch for reasoning over $\mathbf{u}_l(x, y)$. This is in sharp contrast to the single-branch architecture of the transformer models (Figure 3 in Appendix A.2). We further note that when FOLNet is only loaded with j-operator and a-operator, it degenerates into a dual-branch variant of the transformer architecture. In our experiments, we will show that, even in such degenerated setting, FOLNet still outperforms transformer. This is probably because the binary predicate branch explicitly maintains the pairwise relations $\mathbf{u}_l(x, y)$ throughout the reasoning process. In addition, the explicit binary predicate branch also allows us to directly input the relative distance knowledge into the model without performing the less-intuitive operations as in existing RPEs (Shaw et al., 2018). In our experiments in Section 4, we will demonstrate the advantage of such a simple yet effective approach for consuming the relative positional information, along with some other advantages of the dual-branch architecture of FOLNet.

## 4 EXPERIMENTS

### 4.1 EXPERIMENTAL SETTINGS

We now evaluate our FOLNet models under different settings and seek to answer the following question: *Can the neural architecture (FOLNet) that encodes the logical inductive bias learn better language representations than the transformer models?* To this end, we need to eliminate other

| | Model | Params | PE | $D_2$ | Loss | MNLI-m/mm | QQP | QNLI | SST-2 | CoLA | STS-B | MRPC | RTE | Avg |
|---|---|---|---|---|---|---|---|---|---|---|---|---|---|---|
| 1 | BERT | 110M | APE | - | MLM.NSP | 84.5/- | 91.3 | 91.7 | 93.2 | 58.9 | 89.5 | 87.3 | 68.6 | 83.1 |
| 2 | BERT (ours) | 110M | APE | - | MLM.NSP | 83.9/84.1 | 90.9 | 88.2 | 92.6 | 61.5 | 89.2 | 88.2 | 66.8 | 82.7 |
| 3 | FOLNet: j.a | 110M | APE | 12 | MLM.NSP | 84.9/84.5 | 91.1 | 91.6 | 92.2 | 61.1 | 89.6 | 89.5 | 72.2 | 84.0 |
| 4 | FOLNet: j.a | 109M | RPE | 12 | MLM.NSP | 85.0/84.9 | 91.4 | 91.4 | 93.5 | 63.8 | 90.1 | 89.9 | 72.9 | 84.7 |
| 5 | FOLNet: j.a | 110M | RPE$^\star$ | 12 | MLM.NSP | 84.7/84.9 | 91.4 | 91.5 | 93.8 | 63.4 | 89.8 | 90.6 | 72.6 | 84.7 |
| 6 | FOLNet: jm.ap | 123M | RPE | 12 | MLM.NSP | **85.7/85.3** | **91.6** | **91.8** | 93.8 | 65.7 | 90.4 | **91.0** | 73.5 | **85.4** |
| 7 | FOLNet: j.a | 109M | RPE | 16 | MLM.NSP | 85.2/85.2 | 91.3 | 91.8 | 93.7 | 64.1 | 89.9 | 89.3 | 71.8 | 84.6 |
| 8 | FOLNet: j.a | 109M | RPE | 32 | MLM.NSP | **85.8/85.7** | 91.4 | **92.0** | 93.7 | **64.2** | 90.0 | **90.5** | 71.8 | 84.9 |
| 9 | FOLNet: j.a | 110M | RPE | 64 | MLM.NSP | 85.7/85.5 | **91.4** | 91.8 | 93.2 | 63.5 | **90.1** | 90.2 | **74.7** | **85.1** |
| 10 | FOLNet: j.at | 110M | RPE | 64 | MLM.NSP | 86.7/86.6 | 91.6 | 92.8 | 93.1 | 63.6 | 91.1 | 89.9 | 80.9 | 86.2 |
| 11 | FOLNet: j.atp | 117M | RPE | 64 | MLM.NSP | 87.4/87.4 | 91.9 | 93.3 | 94.0 | 62.9 | 91.3 | 91.4 | 81.6 | 86.7 |
| 12 | FOLNet: jm.atp | 124M | RPE | 64 | MLM.NSP | 88.1/87.6 | 91.7 | 93.9 | 94.2 | 64.7 | 91.2 | 91.4 | 83.2 | 87.3 |
| 13 | FOLNet: jmc.atp | 138M | RPE | 64 | MLM.NSP | **88.2/87.9** | **91.9** | **94.1** | **94.5** | **66.9** | **91.6** | **91.5** | **83.5** | **87.7** |
| 14 | FOLNet: jmc.atp | 137M | RPE | 12 | MLM.NSP | 85.9/86.3 | 91.6 | 92.7 | 93.6 | 63.4 | 90.5 | 91.0 | 75.8 | 85.6 |
| 15 | FOLNet: jmc.atp | 138M | RPE | 64 | MLM.SOP | 88.3/87.9 | 91.8 | 94.2 | 94.7 | 65.6 | 91.1 | 91.1 | 83.2 | 87.5 |

Table 3: Analysis of FOLNet on the development sets of GLUE benchmark. All the results are medians of five random seeds. From top to bottom, the first block shows the advantage of dual-branch architecture and compares our new positional encoding to others, the second block analyzes the influence of the binary predicate dimension, the third block performs ablation study of all the logic operators, and the last block shows the results of other pretraining losses for FOLNet. APE stands for absolute positional encoding, RPE$^\star$ denotes the relative positional encoding used by T5 (Shaw et al., 2018; Raffel et al., 2020), and RPE means our proposed relative positional encoding. We have also pretrained a BERT (base) model (line #2) by using the same settings as FOLNet for a fair comparison.

confouding factors and make sure the only difference lies in the model architecture itself. First, we choose to pretrain our FOLNet model using the widely used masked language modeling (MLM) loss (Devlin et al., 2018), and add an extra loss of either the next sentence prediction (NSP) (Devlin et al., 2018) or sentence-order prediction (SOP) (Lan et al., 2019). Many different variants of widely used encoder-only transformer models such as BERT, RoBERTa, ALBERT, DeBERTa and Megatron-LM are pretrained with these losses. Therefore, we will also use these models as our primary baselines. Although there could be other more efficient pretraining losses such as the ones in (Bao et al., 2020; Clark et al., 2020; Yang et al., 2019; Meng et al., 2021), we believe that developing a new model architecture with a better inductive bias is an orthogonal line of research. Therefore, we leave the exploration of other pretraining loss for FOLNet as a future work. In addition, we consider two settings of pretraining dataset: (i) Wikipedia + BookCorpus (Zhu et al., 2015) (16GB in texts) and (ii) a larger set of 160GB texts consisting of Wikipedia, BookCorpus2, OpenWebText2, and Pile-CC (extracted from the Pile dataset (Gao et al., 2020)). We use the BERT tokenizer with 32,768 uncased BPE vocabulary (Sennrich et al., 2016) throughout our experiments.[3] We consider FOLNet models of two different sizes: FOLNet$_{\text{Base}}$ and FOLNet$_{\text{Large}}$, which are comparable in size to the *base* (e.g., BERT$_{\text{Base}}$) and *large* models (e.g., BERT$_{\text{Large}}$) in literature. Finally, the FOLNet model will always be pretrained with a sequence length of 128 tokens, although it will be evaluated on different downstream tasks with longer sequence lengths (e.g., 384 or 512). For evaluation, we consider three benchmarks: GLUE (Wang et al., 2019), SQuAD 2.0 (Rajpurkar et al., 2016b), and FOLIO (Han et al., 2022), where we finetune our pretrained models on each individual task separately for evaluation. More details about these downstream tasks and hyper-parameters can be found in Appendix B.

## 4.2 ANALYSIS OF THE FOLNET ARCHITECTURE

We begin with in-depth analysis of FOLNet to demonstrate its advantage over the transformer architecture. To this end, we first pretrain our FOLNet$_{\text{Base}}$ model on the dataset of Wikipedia and BookCorpus, which is the same as the one used by BERT. And we further use MLM and NSP as our pretraining losses to make it consistent with BERT. We analyze FOLNet from different aspects on GLUE benchmark and report the results in Table 3. We now proceed to discuss the results below.

---

[3]Although Liu et al. (2019) pointed out that it would be more ideal to use the byte-level BPE for preprocessing the much larger 160GB texts, we use the same BERT tokenizer (based on character-level BPE) to process the 160GB text to simplify our logistics, and it already demonstrates the strong performance of our models. Using the byte-level BPE tokenizer with a larger 50K vocabulary as in (Liu et al., 2019; Radford et al., 2019; Brown et al., 2020) may further improve our FOLNet models that are pretrained on the 160GB texts.

| Model | Params | GLUE | | | | | | | | | SQuAD 2.0 | | FOLIO |
|---|---|---|---|---|---|---|---|---|---|---|---|---|---|
| | | MNLI-m/mm | QQP | QNLI | SST-2 | CoLA | STS-B | MRPC | RTE | Avg | EM | F1 | Acc |
| BERT$_{Base}$ | 110M | 84.5/- | 91.3 | 91.7 | 93.2 | 58.9 | 89.5 | 87.3 | 68.6 | 83.1 | 73.7 | 76.3 | 57.8 |
| RoBERTa$_{Base}$ | 110M | 85.8/85.5 | 91.3 | 92.0 | 93.7 | 60.1 | 88.5 | 87.3 | 68.2 | 83.3 | 77.7 | 80.5 | - |
| DeBERTa$_{Base}$ | 134M | 86.3/86.2 | - | - | - | - | - | - | - | - | 79.3 | 82.5 | - |
| FOLNet$_{Base}$ | 138M | **88.2/87.9** | 91.9 | 94.1 | 94.5 | 66.9 | 91.6 | 91.5 | 83.5 | 87.7 | 84.7 | 87.9 | 64.2 |
| BERT$_{Large}$ | 340M | 86.6/- | 91.3 | 92.3 | 93.2 | 60.6 | 90.0 | 88.0 | 70.4 | 84.1 | 79.0 | 81.8 | 62.3 |
| RoBERTa$_{Base}$ | 125M | 87.6/- | 91.9 | 92.8 | 94.8 | 63.6 | 91.2 | 90.2 | 78.7 | 86.4 | 80.5 | 83.7 | **64.7** |
| DeBERTa$_{Base}$ | 134M | 88.8/88.5 | - | - | - | - | - | - | - | - | 83.1 | 86.2 | - |
| FOLNet$_{Base}$ | 138M | **89.4/89.7** | 92.2 | 94.4 | 95.6 | 69.9 | 92.5 | 92.0 | 87.0 | 89.2 | 85.5 | 88.6 | 64.2 |
| RoBERTa$_{Large}$ | 356M | 90.2/90.2 | 92.2 | 94.7 | 96.4 | 68.0 | 92.4 | 90.9 | 86.6 | 88.9 | 86.5 | 89.4 | 67.7 |
| DeBERTa$_{Large}$ | 384M | 91.1/91.1 | 92.3 | 95.3 | 96.8 | 70.5 | - | - | - | - | 88.0 | 90.7 | - |
| ALBERT$_{XXL}$ | 235M | 90.4/- | 92.0 | 95.2 | 96.8 | 68.7 | 92.7 | 90.2 | 88.1 | 89.3 | 87.2 | 89.9 | - |
| ALBERT$_{XXL+}$ | 235M | 90.8/- | 92.2 | 95.3 | **96.9** | 71.4 | **93.0** | 90.9 | 89.2 | 89.9 | 87.4 | 90.2 | - |
| FOLNet$_{Large}$ | 437M | **91.2/91.3** | 92.5 | 95.8 | 96.8 | 71.5 | 92.2 | 93.5 | 91.1 | 90.6 | 88.5 | 91.5 | 70.6 |
| Megatron$_{1.3B}$ | 1.3B | 90.9/91.0 | 92.6 | - | - | - | - | - | - | - | 87.1 | 90.2 | - |
| Megatron$_{3.9B}$ | 3.9B | 91.4/91.4 | 92.7 | - | - | - | - | - | - | - | 88.5 | 91.2 | - |

Table 4: Overall results on the development sets of GLUE, SQuAD 2.0 and FOLIO. The upper block (separated by the solid line) of the table shows the results of the models pretrained on Wikipedia + BookCorpus (16GB), and the lower block are the models pretrained on extended data (160GB). We use dashed lines to separate models of different sizes within each block. All the results are medians of five random seeds. The baseline results of FOLIO are provided by the authors of Han et al. (2022). Here we use ALBERT$_{XXL}$ to refer to the ALBERT model pretrained by 1M steps and use ALBERT$_{XXL+}$ to refer to the ALBERT$_{XXL}$ modeled pretrained by 1.5M steps.

**The advantage of the dual-branch architecture** Recall that when FOLNet only has the join and assoc operators, it can be viewed as a dual-branch version of the transformer architecture. To have a fair comparison, we pretrain a FOLNet model with the same absolute positional encoding (APE) as BERT (line #3 of Table 3). Note that, even in such overly degenerated case, FOLNet still noticeably outperforms BERT on average. When equipping FOLNet with our new relative positional encoding (RPE) (line #4 of Table 3), we will outperform BERT by 2 points on average. Notably, it achieves on par (or slightly better) average performance compared to the one with T5 relative positional encoding (RPE$^\star$ in line #5). As we discussed in earlier section, the T5 RPE are degenerated version of our mu and prod operators. Line #6 of Table 3 show that adding mu and prod operators to the FOLNet would further boost the performance by a noticeable amount.

**The benefits of a larger $D_2$** We can see (lines #7-9 of Table 3) that increasing the dimension $D_2$ will steadily improves the performance. As we will reveal soon, when FOLNet is fully loaded with all the operators, having a larger $D_2$ is essential to unleash their full power; that is, the performance improvement from increasing $D_2$ would be even larger for a fully-loaded FOLNet.

**Contribution of the logic operators** We now analyze the contributions of the logic operators by adding them one-by-one into FOLNet until being fully loaded. We see from line #9 to line #13 in Table 3 that this drastically improves the average performance. In line #14, we evaluate a fully-loaded FOLNet with $D_2$ decreased from $64$ to $12$, which shows a significant performance drop. This confirms the importance of having a relatively large $D_2$ in order to store the deduced relations.

## 4.3 OVERALL PERFORMANCE

Having closely examined various aspects of FOLNet architecture, we now proceed to evaluate it comprehensively on three benchmarks (GLUE, SQuAD 2.0, and FOLIO). We will also examine its performance when we further scale up the pretraining data size and model size. To pretrain the model on a larger (160GB) dataset, we find it more efficient to generate the pretraining data with SOP losses. This is because NSP losses require us to sample negative sentences from another document. To begin with, we verify the performance by pretraining a FOLNet$_{Base}$ with SOP loss on Wikipedia and BookCorpus. The result (line #15 in Table 3) shows that it could slightly degrade the performance compared to the one with NSP (line #13). However, this is relatively tolerable given its convenience when pretraining on a large corpus. Therefore, we will replace NSP with SOP when pretraining

FOLNet$_\text{Base}$ and FOLNet$_\text{Large}$ on the 160GB dataset. We show our results on the GLUE benchmark in Table 4 and compare them with other baseline methods. Observe that our FOLNet$_\text{Base}$ models pretrained on both 16GB data and 160GB significantly outperform other transformer-based models on all three benchmarks. Our FOLNet$_\text{Base}$ pretrained on 16GB data outperforms BERT$_\text{Large}$ model that is $3\times$ larger in model size. In addition, it even outperforms RoBERTa$_\text{Base}$ that is pretrained on 160GB data. When pretraining our FOLNet$_\text{Base}$ model on 160G data, we even surpass the RoBERTa$_\text{Large}$ model (89.3 vs 88.9) on GLUE benchmark. Likewise, our FOLNet$_\text{Large}$ model also significantly outperforms all other baselines. It even outperforms ALBERT$_\text{XXL+}$ that is pretrained by 1.5M steps (i.e., 50% more tokens during pretraining). Notably, our FOLNet$_\text{Large}$ model achieves comparable performance as the Megatron$_\text{3.9B}$ model on both GLUE and SQuAD 2.0 benchmark, which is 10 times larger than our model. Furthermore, although our FOLNet model is not designed for solving reasoning problem (but incorporating reasoning as an inductive bias at the token-level), our model still consistently demonstrates stronger first-order logic reasoning capability on FOLIO task (e.g., $+3.9$ over RoBERTa$_\text{Large}$). Finally, we would like to highlight that all the FOLNet$_\text{Base}$ and FOLNet$_\text{Large}$ models are pretrained with sequence length of 128, in contrast to 512 as in other baselines. However, they are evaluated on GLUE, SQuAD 2.0 and FOLIO with sequence length of 128, 384 and 512, respectively. In particular, by finetuning with merely 1,004 training examples on FOLIO, it is able to generalize to much longer sequences (512), which have never been seen during pretraining.

## 5 Related Works

**Transformer language models** There have been a long line of research on neural language models since Bengio et al. (2000). Recently, it has achieved great success by exploring different variants of pretrained transformer models (Vaswani et al., 2017) for solving downstream language tasks, such as with finetuning (Radford et al., 2018; Devlin et al., 2018; Lan et al., 2019; Liu et al., 2019; Yang et al., 2019) or with zero/few-shot learning using large language models (Radford et al., 2019; Brown et al., 2020; Chowdhery et al., 2022). Another line of active research focuses on developing more effective pretraining losses (Yang et al., 2019; Clark et al., 2020; Bao et al., 2020; Tay et al., 2022) beyond the widely used autoregressive or masked language modeling objectives. There have been limited works on developing new neural architectures for learning better language representations. In this paper, we seek to move in this direction and develop a new neural architecture based on logical inductive bias.

**Logic programming and neural reasoning** Our logical inductive bias is inspired by logic programmings (Horn, 1951; De Raedt et al., 2007; De Raedt & Kimmig, 2015; Fierens et al., 2012; Manhaeve et al., 2018) and inductive logic programming (Evans & Grefenstette, 2018; Muggleton, 1991; 1996; Friedman et al., 1999). Different from these works, we do not directly work on reasoning problems. Instead, we seek to encode the logical inductive bias into the neural model to learn better language representations. Another line of related works focuses on developing neural models that can perform reasoning in a broad sense. For example, different variants of memory augmented networks are developed (Le et al., 2020; Santoro et al., 2018; Graves et al., 2014; 2016; Santoro et al., 2016; Le et al., 2019), which augment a control network with memory units and the associated neural read/write modules. Besides, Rocktäschel & Riedel (2017) and Minervini et al. (2020) consider the problem of proving structured queries to knowledge base, by constructing differentiable neural networks via backward-chaining. Bergen et al. (2021) develop a triangular attention to deduce relations from other relations, which can be viewed as a transformer with a single relational branch. These methods are effective in solving (relatively small-scale) reasoning tasks. However, it remains unclear whether they can be effectively pretrained on large-scale texts to solve diverse downstream natural language tasks.

## 6 Conclusion

We introduce a novel logical inductive bias, which treats language representation learning as a logic programming problem. We then develop a fully-differentiable neural architecture (FOLNet) that effectively encodes this inductive bias by forward-chaining a rich set of neural logic operators. The proposed FOLNet architecture has the same input-output interface as the transformer models and can be pretrained over large-scale text data. Experimental results demonstrate that the FOLNet architecture significantly outperforms different variants of transformer models, and has many inherent advantages due to the new dual-branch architecture along with its rich set of neural logic operators.

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

# Supplementary Materials

## A    ADDITIONAL DETAILS OF THE FOLNET ARCHITECTURE

### A.1    MORE DISCUSSIONS ON THE NEURAL LOGIC OPERATORS

**Modus Ponens inference with restricted kernels.**    First, it is straightforward to show that the neural logic operators in Table 2 can be implemented as batch matrix multiplications (Figure 2), where multiple slices of matrix multiplications are executed in parallel to obtain the outputs. Different operators pick their own reduction dimensions (R-dim) and batching dimensions (B-dim) for the matrix multiplication (see Table 2). For example, the join operator picks the $a$-dimension as the R-dim and the $h$-dimension as the B-dim so that matrix multiplications are carried out in parallel over $h$. According to (9), each slice of the matrix multiplication can be viewed as an independent neural Modus Ponens inference based on its own set of clauses. For example, in the join operator, let $\mathbf{K}_{\text{join},h}$ denote a matrix that collects $\mathbf{K}_h(x,a)$ as its $(x,a)$-th element. Then, $\mathbf{K}_{\text{join},h}$ is the $h$-th kernel slice that represents a particular group of clauses for the join operator. In addition, recall that the values at each row of the kernel slice characterize the conjunction pattern on the right-hand side of (8), with positive (or negative) values determine whether the original premise $P_m$ (or the negated premise $\neg P_m$) should be used for conjunction (see Section 3.2 and Appendix D). Therefore, the R-dim in the matrix multiplication shall be understood as the conjunction dimension in (8). In the join operator, it implies that the conjunction operation is over the $a$-dimension of the premises $\mathbf{v}_{hs}(a)$, with the conjunction pattern determined by the $x$-th row of $\mathbf{K}_{\text{join},h}$ if we want to infer $\mathbf{u}_{hs}(x)$. In contrast, a full Modus Ponens infernece from unary atoms to unary atoms (the first expression in (7)) shall perform its conjunction over the joint dimensions of $(h,s,a)$, which is much higher in complexity. Therefore, the join operator controls the complexity of Modus Ponens inference by restricting the conjunction operation over the $a$-dimension. Furthermore, it is noteworthy that the join operator is applying the same kernel slice to premises of different $s$; that is, it implements *kernel-sharing* across the $s$-dimension. This is another complexity-reduction strategy resulted from the principles in Section 3.2. Likewise, the same conclusion holds for all other operators, who pick their own conjunction-dimensions and kernel-sharing dimensions (see Figure 2).

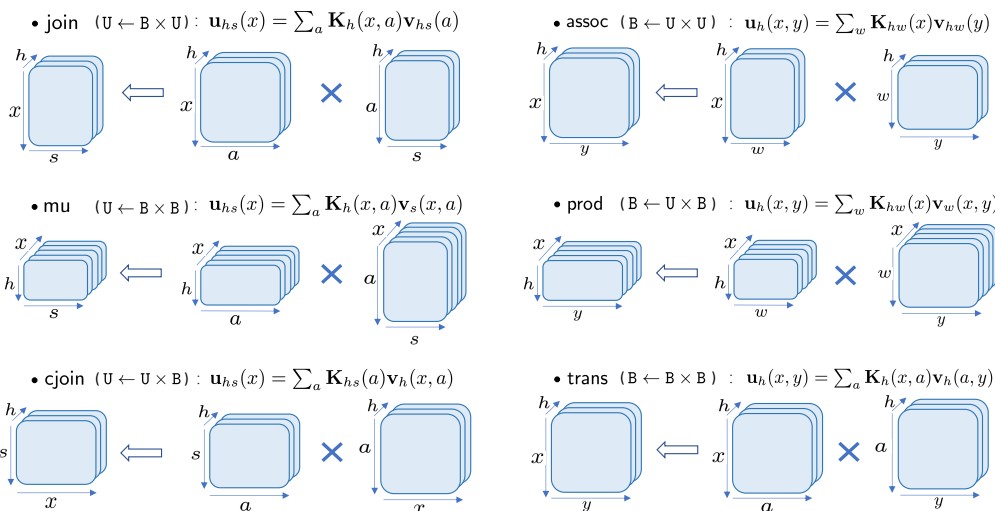

Figure 2: Our neural logic operators can be efficiently implemented as batch matrix multiplications by using hardware accelerators (e.g., GPUs and TPUs). Each slice of matrix multiplications represents an independent neural Modus Ponens inference based on its own set of clauses.

**Naming protocols and examples.**    We now explain the naming protocols for the logic operators along with concrete examples to demonstrate how each of them help reasoning in different aspects.

- bool: It operates over the predicate dimension and plays the role of Boolean functions over a set of input truth values, which leads to its name of bool-operator. It is used to deduce a property of $x$ from other properties regarding the same $x$ via certain Boolean operations. For example, it can model inferences like $\text{Edible}(x) \leftarrow \text{IsMushroom}(x) \wedge \neg\text{IsToxic}(x)$, where the conjunction pattern on the right-hand side is determined by the kernel.

- join: We name this operator as join-operator because it is in a similar form as the join operation in $\lambda$-DCS (Liang, 2013). For example, when the kernel represents $\text{IsBornIn}(x, y)$, it can be used to infer $\text{CitizenOfUSA(x)}$ from premise $\text{CountryIsUSA}(y)$, where the kernel determines the conjunction pattern of $\text{CountryIsUSA}(y)$ over $y$.

- cjoin: Since it simply swaps the roles of the kernel and the premise in the join-operator, we name it as the conjugate join operator. It plays a similar role as the join operator.

- mu: We name it as mu-operator because it is can be viewed as a more general-form of the mu-abstraction operation in $\lambda$-DCS (Liang, 2013). For example, when the kernel models $\text{ApplyJobAt}(x, y)$, it can deduce $\text{HasAJob}(x)$ from premise $\text{ReceiveOfferFrom}(x, y)$, where the kernel determines the conjunction pattern of $\text{ReceiveOfferFrom}(x, y)$ over $y$.

- assoc: We name it as assoc-operator because it can be viewed as computing the association between two vectors. For example, when the kernel represents $\text{LosAngeles}(y)$, it can be used to infer $\text{SameStateAs}(x, y)$ from $\text{SanFrancisco}(x)$, where the kernel determines the conjunction pattern of $\text{SanFrancisco}(x)$ over the predicate dimension. In this example, we only have one premise atom over the predicate dimension. In the general case, a particular conjunction pattern would be applied to multiple premise atoms to yield the output.

- prod: We name it as prod-operator because it is in resemblance of computing the product over the predicate dimension. For example, when the kernel models $\text{Graduated}(x)$, it can be used to infer $\text{GraduatedFrom(x, y)}$ from $\text{StudyAt(x, y)}$, where the kernel determines the conjunction pattern of $\text{StudyAt(x, y)}$ over the predicate dimension. In this example, we only have one premise atom over the predicate dimension. In the general case, a particular conjunction pattern would be applied to multiple premise atoms to yield the output.

- trans: We name it as trans-operator because its form is in reminiscent of reasoning with transitive properties. For example, when the kernel models $\text{FatherOf}(x, z)$, it can be used to infer $\text{GrandFatherOf}(x, y)$ from $\text{ParentOf}(z, y)$, where the kernel determines the conjunction pattern of $\text{ParentOf}(z, y)$ over the dimension $z$.

Note that we do not have a logic operator corresponding to the typing $\text{B} \leftarrow \text{B} \times \text{U}$ because it will be identical to the prod operator when the kernel action function is chosen to be the identity mapping. In addition, the above examples further show that the kernels (i.e., the clauses) themselves are also atoms or can be computed from certain atoms. This justifies our strategy of parameterizing the kernels by (the intermediate deduction results of) FOLNet itself in Section 3.3.

## A.2    COMPARING TO THE TRANSFORMER ARCHITECTURE

Figure (3) show that the transformer architecture can be understood as a single-branch version of the FOLNet architecture with only join-operator and assoc-operator.

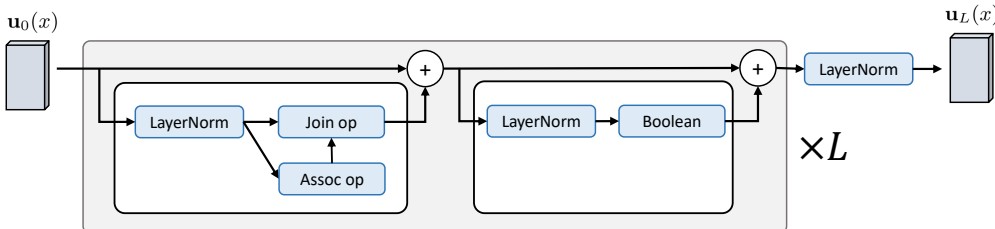

Figure 3: Overview of the Transformer architecture. It can be understood as a single-branch version of the FOLNet architecture with only join-operator and assoc-operator.

### A.3 THE BOOLEAN OPERATOR

In FOLNet, we do not chain the Boolean operators in parallel with other operators but chain it with others in a cascade manner. This is akin to cascading the FFN with the self-attention module in transformers. Nevertheless, we may also place it in parallel to other operators just as making FFN in parallel to self-attention in transformer, which is done in Chowdhery et al. (2022). In addition, we also adopt the same FFN architecture (i.e., a multilayer perceptron with GeLU units) for the Boolean operator in order to make FOLNet directly comparable with transformer architectures.

### A.4 THE INPUT-OUTPUT INTERFACE

**Computing the pretraining losses (e.g., MLM, NSP and SOP) from the output atoms**   As we pointed out, FOLNet models will have a similar input-output interface as the existing transformer models, so that we can seamless adopt existing pretraining losses (e.g., MLM, NSP and SOP) by computing them from $\{\mathbf{u}_L(x)\}$. Recall that $\mathbf{u}_L(x_t)$ is the vector that represents the derived (advanced) properties for the object (token) $x_t$, where $t = 1, \ldots, T$. For a masked token $x_t$, $\mathbf{u}_L(x_t)$ contains its properties that could be deduced from other tokens via their input properties and relations. Therefore, these deduced properties in $\mathbf{u}_L(x_t)$ can be used to predict the original masked token. For example, we can apply a linear classifier followed by a softmax operator to compute a probability distribution over the vocabulary and the MLM loss. Likewise, to compute NSP and SOP losses, we add a special "[CLS]" token at the beginning of the input sequence, so that the $\mathbf{u}_L(x)$ that corresponds to the "[CLS]" token will be fed into a binary classifier for computing the NSP or SOP loss. The usage of $\{\mathbf{u}_L(x_t)\}$ in downstream tasks (such as sequence classification, multiple-choice and sequence labeling) are also similar. On the other hand, we have not used the output binary predicates $\{\mathbf{u}_L(x_t, y_\tau)\}$ for computing any losses. The main reason is that we would like to adopt the existing off-the-shelf pretraining losses for an apple-to-apple comparison regarding the proposed model architecture. Since most of these losses are developed for the transformer architecture, which is a single-branch model with only unary predicates on its main pathway (Figure 3), it is not surprising that these losses are mainly computed from the unary properties $\{\mathbf{u}_L(x_t)\}$. Nevertheless, we believe that our newly introduced binary predicate branch could open a new avenue for developing additional pretraining losses using $\mathbf{u}_L(x_t, y_\tau)$ (i.e., the token relations). For example, we may randomly swap two tokens $x_t$ and $x_\tau$ and use $\mathbf{u}_L(x_t, x_\tau)$ to predict whether they have been swapped or not. We will leave the development of more effective pretraining losses for FOLNet as a future work.

**The input base atoms in $\mathcal{B}$ and their logit vector representations**   Throughout our work, we only consider plain texts as the input, which is similar to other pretrained language models. Therefore, the base atoms at the input should only encode the information that is directly available from the plain text, which include the token ids in the input sequence (denoted by $\texttt{TokenID}_v(x_t)$) and the relative distance between any of the two tokens (denoted by $\texttt{RelDist}_d(x_t, x_\tau)$). Specifically, $\texttt{TokenID}_v(x_t) = \mathsf{T}$ if the token $x_t$ takes the $v$-th id in the vocabulary for $v = 1, \ldots, V$, and $\texttt{TokenID}_v(x_t) = \mathsf{F}$ otherwise. When the input sequence consists of a pair of sequences, we will construct the input sequence as:

$$\text{``[CLS] Sequence \#0 [SEP] Sequence \#1 [SEP] [PAD] \ldots [PAD]''},$$

which is similar to the input format used in BERT and other transformer models. In this case, we will introduce an additional base atom $\texttt{SeqID}_s(x_t)$ to characterize whether a token $x_t$ belongs to the $s$-th sequence ($s = 0, 1$); that is, $\texttt{SeqID}_s(x_t) = \mathsf{T}$ if token $x_t$ belongs to the $s$-th sequence and is $\mathsf{F}$ otherwise. This atom is akin to the segment ids (or token type ids) in existing pretrained transformer models. Furthermore, the relative positional atom $\texttt{RelDist}_d(x_t, x_\tau) = \mathsf{T}$ if $d = \text{dist}_\Delta(x_t, x_\tau)$ and is $\mathsf{F}$ otherwise, where $\text{dist}_\Delta(x_t, x_\tau)$ is a clipped distance function with a clipping parameter $\Delta$:

$$\text{dist}_\Delta(x_t, x_\tau) = \begin{cases} \max(1-\Delta, \min(\tau-t, \Delta-1)) & t > 0, \tau > 0 \text{ and } \texttt{SeqID}_s(x_t) = \texttt{SeqID}_s(x_\tau) \\ \Delta + 1 & t > 0, \tau > 0 \text{ and } \texttt{SeqID}_s(x_t) \neq \texttt{SeqID}_s(x_\tau) \\ \Delta & t = 0, \tau > 0 \\ -\Delta & t > 0, \tau = 0 \\ 0 & t = 0, \tau = 0 \end{cases}.$$

The above clipped distance function clamps the relative distance to be within $[-\Delta + 1, \Delta - 1]$ when the two tokens are from the same sequence. And it also assigns a special distance id whenever they

belong to two different sequences. In other words, the distance between any two tokens would be the same if they belong to two separate sequences. Likewise, we also characterize the distances between the [CLS] token ($t = 0$ or $\tau = 0$) and the other tokens ($\tau > 0$ or $t > 0$) with two special distance ids, which sets all the regular tokens to have the same distance towards (or from) the special [CLS] token. Such relative positional encoding (atom) preserves the translational invariance of the text sequence and generalizes better than the absolute positional encodings used in BERT and RoBERTa. Notably, it allows FOLNet to generalize to much longer sequences that are unseen during pretraining. In summary, we consider the following base atoms as the inputs to the FOLNet model:

$$\mathcal{B} = \{\texttt{TokenID}_v(x_t), \texttt{SeqID}_s(x_t), \texttt{RelDist}_d(x_t, x_\tau) | t, \tau = 1, \ldots, T\}. \tag{10}$$

By stacking $\texttt{TokenID}_v(x_t)$ and $\texttt{SeqID}_s(x_t)$ over $v$ and $s$, stacking $\texttt{RelDist}_d(x_t, x_\tau)$ over $d$, and using the values of 1 and 0 to represent $\mathsf{T}$ and $\mathsf{F}$, respectively, we will have the 0-1 vectors to represent all the input atoms in $\mathcal{B}$. Then, we may further convert them into dense vector representations $\{\mathbf{u}_0(x_t), \mathbf{u}_0(x_t, x_\tau)\}_{t,\tau}$ via embedding lookup. The embedding lookup process can also be understood as finding a set of more fine-grained learnable properties (or relations) to characterize a given token $x_t$ (or a token-pair $(x_t, x_\tau)$), where these properties (or relations) are represented in the same *logit space* as the neural logic operators in Table 2, which carry out Modus Ponens inferences. Finally, the above base atoms are just one particular design for our FOLNet model, and there could be other alternatives for encoding the input information. And when there is extra information available besides plain texts, we may also encode it as the new unary or binary base atoms in $\mathcal{B}$.

## A.5 EXTENDING FOLNET TO TEXT-TO-TEXT VERSIONS

So far, we have mainly focused on the encoder-only version of the FOLNet model. We now show that it can be extended to the decoder-only and the encoder-decoder counterparts in a relatively straightforward manner. Such extension would be useful for text-to-text generation tasks.

**Decoder-only**   To develop the decoder-only version of FOLNet, we need to let the model auto-regressively generate the output tokens. In the context of FOLNet, this requires the unary and binary atoms $\{\mathbf{u}(x_t), \mathbf{u}(x_t, x_\tau)\}$ to be inferred only from the premises in the past: $\{\mathbf{v}(x_\omega), \mathbf{v}(x_\omega, x_\nu) : \omega \leq t\}$. In addition, we further restrict the binary atoms to have a causal pattern, i.e., $\mathbf{u}(x_t, x_\tau) = 0$ and $\mathbf{u}(x_\omega, x_\nu) = 0$, whenever $t < \tau$ and $\omega < \nu$. Figure 4(a) illustrates these patterns for the unary and the binary atoms. Note that the causal structure for the binary atoms translates into a lower triangular pattern for the nonzeros. In particular, the dark blue atoms are directly responsible for the generation of the next token (word), and the auto-regressive generation requires the model to update them only from the light blue atoms. We enforce such auto-regressive property by multiplying a proper 0-1 mask to the binary kernels and premises in the neural logic operators. In the last column of Table 5, we list the kernels and the premises that should be masked, and in Figure 4(b), we show the masks that correspond to two variants of decoder-only models: (i) Causal Langauge Model (CausalLM) (Radford et al., 2018; 2019; Brown et al., 2020; Chowdhery et al., 2022), and (ii) Prefix Langauge Model (PrefixLM) (Liu et al., 2018; Raffel et al., 2020). The CausalLM has a lower triangular mask for the binary atoms so that the information pattern is always unidirectional. On the other hand, the PrefixLM will have a bi-directional mask for the input segment (the top-left part in Figure 4(b)) and a unidirectional mask for the output (target) segment (the bottom-right part in Figure 4(b)). Accordingly, the binary atoms of the PrefixLM version will share a similar pattern as its mask in Figure 4(b), which models the relatons for the prefix and the output segments separately. With such simple modifications, our decoder-only FOLNet model will have the same input-output interface as the decoder-only transformers; it can be pretrained to predict the next tokens using a linear classifier over the unary atoms $\mathbf{u}_L(x_t)$. After the training, it can generate tokens in an auto-regressive manner.

**Encoder-Decoder**   For the encoder-decoder variant, we use two separate stacks of FOLNet for the encoder and decoder, where each of them has the same overall architecture as in Figure 1 with its own set of atoms (the green and blue blocks in Figure 4(c)). In particular, the encoder will be identical to the encoder-only version that we have thoroughly discussed earlier in the main paper. Meanwhile, the decoder part will be similar to the decoder-only variant with a few additional modifications. First, the decoder needs to maintain a slightly different version of binary atoms (Figure 4(c)). Specifically, besides the relations between the output tokens, the decoder also has to model the (unidirectional) relations from the input tokens to the output tokens (i.e., the bottom-left part of $\mathbf{u}_h(x, y)$ in Figure 4(c)). These relations are crucial in deducing the output tokens from the input ones, which plays a

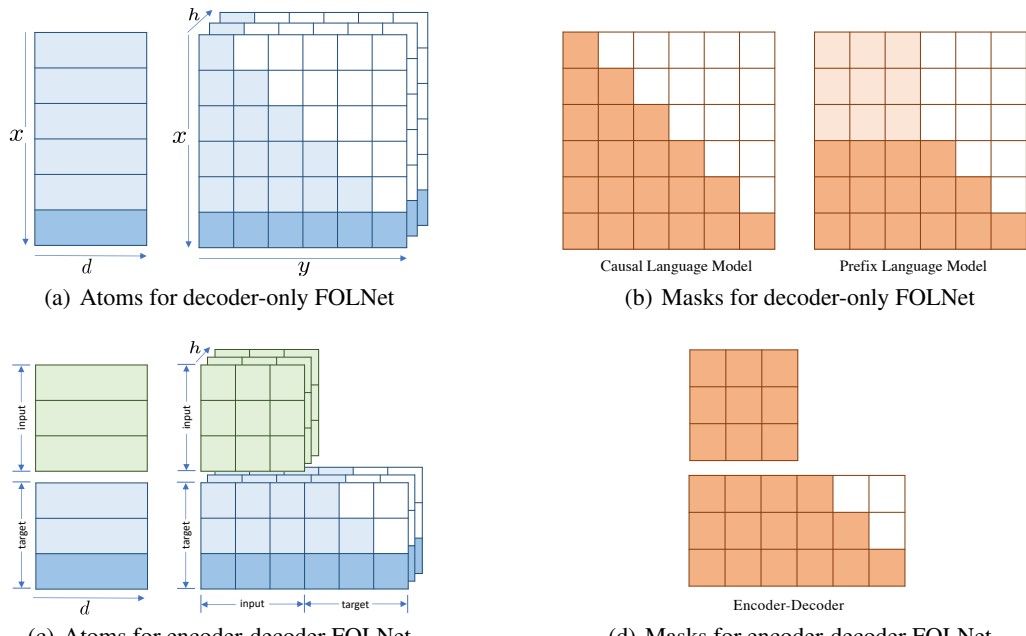

(a) Atoms for decoder-only FOLNet

(b) Masks for decoder-only FOLNet

(c) Atoms for encoder-decoder FOLNet

(d) Masks for encoder-decoder FOLNet

Figure 4: Extending FOLNet to text-to-text versions. The top row illustrates the atoms and the masking strategy for the decoder-only version, and the bottom row shows the encoder-decoder variant. On the left side, we visualize the unary and the binary atoms $\{\mathbf{u}_d(x), \mathbf{u}_h(x,y)\}$ for these cases, where $d = (h-1)S + s$. The green and blue colors characterize the nonzero patterns for the atoms of the encoder and the decoder, respectively. The green-colored atoms are associated with the encoder part in the encoder-decoder variant. The dark blue color represents the atoms that are directly responsible for generating the next token in the decoder. Notably, they are deduced only from the light blue atoms, so that the entire generation process retains an auto-regressive nature. On the right side, the white and the orange blocks represent the 0-1 masking positions, where the light orange part are associated with the prefix segment in PrefixLM. Likewise, the encoder-decoder variant has separate sets of masks for the encoder and the decoder, respectively, where the mask for the encoder part are designed to have a bi-directional information pattern. Similar design also holds for the prefix part in PrefixLM.

| Sym. | Ops. | Typing | B-dim. | R-dim. | Neural operator in logit space | Mask |
|------|------|--------|--------|--------|-------------------------------|------|
| b | bool | $U \leftarrow U \times U$ | $x$ | $w$ | $\mathbf{u}_{hs}(x) = \sum_w \mathbf{K}_{hw}(x)\mathbf{v}_{ws}(x)$ | - |
| c | cjoin | $U \leftarrow U \times B$ | $h$ | $a$ | $\mathbf{u}_{hs}(x) = \sum_a \mathbf{K}_{hs}(a)\mathbf{v}_h(x,a)$ | $\mathbf{v}_h(x,a)$ |
| j | join | $U \leftarrow B \times U$ | $h$ | $a$ | $\mathbf{u}_{hs}(x) = \sum_a \mathbf{K}_h(x,a)\mathbf{v}_{hs}(a)$ | $\mathbf{K}_h(x,a)$ |
| m | mu | $U \leftarrow B \times B$ | $x$ | $a$ | $\mathbf{u}_{hs}(x) = \sum_a \mathbf{K}_h(x,a)\mathbf{v}_s(x,a)$ | $\mathbf{K}_h(x,a), \mathbf{v}_s(x,a)$ |
| a | assoc | $B \leftarrow U \times U$ | $h$ | $w$ | $\mathbf{u}_h(x,y) = \sum_w \mathbf{K}_{hw}(x)\mathbf{v}_{hw}(y)$ | - |
| p | prod | $B \leftarrow U \times B$ | $x$ | $w$ | $\mathbf{u}_h(x,y) = \sum_w \mathbf{K}_{hw}(x)\mathbf{v}_w(x,y)$ | $\mathbf{v}_w(x,y)$ |
| t | trans | $B \leftarrow B \times B$ | $h$ | $a$ | $\mathbf{u}_h(x,y) = \sum_a \mathbf{K}_h(x,a)\mathbf{v}_h(a,y)$ | $\mathbf{K}_h(x,a), \mathbf{v}_h(a,y)$ |

Table 5: The binary kernels and premises to be masked for decoder-only versions of FOLNet. In the encoder-decoder variant, similar part of the kernels and premises would be masked in its decoder.

similar role as the cross-attention scores in transformers. Accordingly, we also need to adjust the masks to handle these relations separately (see Figure 4(d)). Second, the neural logic operators in Table 5 should also be slightly adjusted in order to incorporate the additional premise atoms from the encoder output. For example, the premise $\mathbf{v}_{hs}(a)$ in join-operator should now be a concatenation of the unary atoms from the encoder output (with a linear projection) and the decoder premises. Likewise, the premise $\mathbf{v}_{hw}(y)$ in assoc-operator should also be a concatenation of the encoder output (with a linear projection) and the decoder premises. These two modified operators share a similar spirit as the self-attention and the cross-attention mechanisms in transformer decoders. The cjoin operator is similar to join operator, except now the concatenation happens at the kernel $\mathbf{K}_{hs}(a)$ (with

a linear projection). The mu-operator and the prod-operator stay the same as before. Meanwhile, the trans-operator for the decoder needs to be implemented according to the following new expression:

$$\mathbf{u}_h(x,y) = \begin{cases} \sum_{a \in \mathcal{T}} \mathbf{K}_h(x,a)\mathbf{v}_h^{\mathrm{D}}(a,y) & y \in \mathcal{T} \\ \sum_{a \in \mathcal{I}} \mathbf{K}_h(x,a)\mathbf{v}_h^{\mathrm{E}}(a,y) + \sum_{a \in \mathcal{T}} \mathbf{K}_h(x,a)\mathbf{v}_h^{\mathrm{D}}(a,y) & y \in \mathcal{I} \end{cases}$$

where $\mathcal{T}$ and $\mathcal{I}$ are defined to be the target and the input sequences, respectively, the kernel $\mathbf{K}_h(x,a)$ are obtained by applying a linear projection to the binary atoms $\mathbf{u}(x,a)$ in the decoder, and the superscript E or D in the premises $\mathbf{v}_h(a,y)$ denotes whether they are from the encoder or the decoder. Notably, we observe that the encoder-decoder version of FOLNet retains the dual-branch architecture in its decoder module as well. This is in sharp contrast to the standard transformer decoder, which is a single-branch architecture with only unary atoms on its main pathway.

# B  EXPERIMENTAL DETAILS

## B.1  OVERVIEW OF THE DOWNSTREAM TASKS

**GLUE**  The GLUE benchmark (Wang et al., 2019) consists of 9 tasks: MNLI (Williams et al., 2018), QQP (Iyer et al., 2017), QNLI (Rajpurkar et al., 2016a), SST-2 (Socher et al., 2013), CoLA (Warstadt et al., 2019), STS-B (Cer et al., 2017), MRPC (Dolan & Brockett, 2005), RTE (Dagan et al., 2005; Haim et al., 2006; Giampiccolo et al., 2007; Bentivogli et al., 2009), WNLI (Levesque et al., 2012). They cover a wide range of natural language understanding tasks such as natural language inference, paraphrasing, linguistic acceptability, and sentiment analysis. We finetune our FOLNet models by following the same procedures from BERT (Devlin et al., 2018). We do not evaluate our model on WNLI because it generally needs special procedures. The basic description of all the tasks in GLUE (including their evaluation metrics) can be found in Table 6.

**SQuAD 2.0**  SQuAD 2.0 (Rajpurkar et al., 2016b) is an extractive question answering dataset built from Wikipedia. The objective of the task is to predict an answer span from the context paragraphs. SQuAD 2.0 is an updated version that adds additional 50,000 unanswerable questions to the original SQuAD 1.1 version. The performance metrics are exact match (EM) and F1 scores.

**FOLIO**  FOLIO (Han et al., 2022) is a natural language reasoning dataset that contains first-order logic reasoning problems. It requires the models to decide whether a conclusion statement is correct or not given a world defined by a set of premises. It is formulated as a 3-class classification problem with the labels being "True", "False", "Unknown". We follow the same procedure as Han et al. (2022) by formulating the problem as a sequence pair classification problem. Specifically, we concatenate all the the premises into one sequence (i.e., sequence A), and then further concatenate it with the conclusion (i.e., sequence B), where the two sequences are separated by a [SEP]. In addition, we add a [CLS] token at the beginning and a [SEP] in the end before padding (in the end). By doing so, the task has the same format as a natural language inference task (e.g., MNLI). Table 7 (quoted from the original FOLIO paper (Han et al., 2022)) shows an example from the FOLIO dataset, which demonstrate that it requires strong first-order reasoning capabilities to solve the problem. The dataset has an official train/validation/test split with 1,004/204/227 examples, respectively. By the time of this submission, the test set is not available and thus we only report performance on the validation set.

| | MNLI | QQP | QNLI | SST-2 | CoLA | RTE | MRPC | STS-B |
|---|---|---|---|---|---|---|---|---|
| **Size** | 393K | 364K | 108K | 67K | 8.5K | 2.5K | 3.7K | 5.7K |
| **Task** | Inference | Similarity | QA/Inference | Sentiment | Acceptability | Inference | Paraphrase | Similarity |
| **Metric(s)** | Accuracy | Accuracy/F1 | Accuracy | Accuracy | Matthews corr. | Accuracy | Accuracy/F1 | Pearson/Spearman. |
| **#Classes** | 3 | 2 | 2 | 2 | 2 | 2 | 2 | 1 (regression) |

Table 6: Basic information about different tasks in GLUE benchmark.

## B.2  IMPLEMENTATION DETAILS AND HYPER-PARAMETERS

We implement both our pretraining and finetuning pipelines using PyTorch (Paszke et al., 2019) and automatic mixed precision (AMP) learning (Micikevicius et al., 2018) based on the Apex library (Nvidia, 2019). For pretraining, we use large-batch training (with a batch-size 131K) using LAMB

| A FOLIO example based on the Wild Turkey Wikipedia page | |
| --- | --- |
| **NL premises** | **NL Conclusion -> Labels** |
| 1. There are six types of wild turkeys: Eastern wild turkey, Osceola wild turkey, Gould's wild turkey, Merriam's wild turkey, Rio Grande wild turkey, and the Ocellated wild turkey. 
 2. Tom is not an Eastern wild turkey. 
 3. Tom is not an Osceola wild turkey. 
 4. Tom is also not a Gould's wild turkey, or a Merriam's wild turkey, or a Rio Grande wild turkey. 
 5. Tom is a wild turkey. | A. Tom is an Ocellated wild turkey. -> True 
 B. Tom is an Eastern wild turkey. -> False 
 C. Joey is a wild turkey. -> Unknown |
| **FOL Premises** | **FOL conclusion -> Labels** |
| 1. $\forall x(\texttt{WildTurkey}(x) \rightarrow (\texttt{Eastern}(x) \vee \texttt{Osceola}(x) \vee \texttt{Goulds}(x)$ 
 $\vee \texttt{Merriams}(x) \vee \texttt{Riogrande}(x) \vee \texttt{Ocellated}(x)))$ 
 2. $\neg(\texttt{WildTurkey}(tom) \wedge \texttt{Eastern}(tom))$ 
 3. $\neg(\texttt{WildTurkey}(tom) \wedge \texttt{Osceola}(tom))$ 
 4. $\texttt{WildTurkey}(tom) \rightarrow \neg(\texttt{Goulds}(tom) \vee \texttt{Merriams}(tom) \vee \texttt{Riogrande}(tom))$ 
 5. $\texttt{WildTurkey}(tom)$ | A. $\texttt{Ocellated}(tom)$ -> True 
 B. $\texttt{Eastern}(tom)$ -> False 
 C. $\texttt{WildTurkey}(joey)$ -> Unknown |

Table 7: We show an example from the FOLIO dataset, which is quoted directly from the original FOLIO paper (Han et al., 2022). It demonstrates that it requires strong first-order reasoning capabilities to solve the problem. "NL" stands for "natural language" and "FOL" stands for "First-Order Logic". We only use the natural language part to solve the task in our experiments.

optimizer (You et al., 2020). The pretraining of FOLNet$_\text{Base}$ on Wikipedia + BookCorpus (16GB) for 8K steps takes about 12 hours using 512 V100 GPUs. The pretraining of FOLNet $_\text{Base}$ on 160GB data for 128K steps takes 7 days using 512 V100 GPUs. And the pretraining of FOLNet$_\text{Large}$ on 160GB data for 128K steps takes 19 days using 512 V100 GPUs. For the finetuning of all downstream tasks, we also use AMP learning based on Apex, and the optimizers are FusedAdam from Apex library.

We report the hyper-parameters of pretraining FOLNet in Table 8. The hypper-parameters for finetuning different downstream tasks are included in Table 9. The hyper-parameters for the finetuning tasks are searched per task, and the results are the median of five random seeds.

| Hyperparams | FOLNet$_\text{Base}$ | FOLNet$_\text{Base}$ | FOLNet$_\text{Large}$ |
| --- | --- | --- | --- |
| Pretraining data size | 16G | 160G | 160G |
| Number of Layers ($L$) | 12 | 12 | 24 |
| Unary Hidden size ($D_1$) | 768 | 768 | 1024 |
| Binary Hidden size ($D_2$) | 64 | 64 | 64 |
| Unary Boolean (FFN) intermediate size | 3072 | 3072 | 4096 |
| Binary Boolean (FFN) intermediate size | 256 | 256 | 256 |
| Number of Attention heads ($H$) | 12 | 12 | 16 |
| Attention head size ($S$) | 64 | 64 | 64 |
| RPE clipping parameter ($\Delta$) | 64 | 64 | 64 |
| Dropout rate | 0.1 | 0.1 | 0.1 |
| Attention Dropout rate | 0.1 | 0.1 | 0.1 |
| Warmup Ratio | 1% | 1% | 1% |
| Peak Learning Rate | 1e-2 | 2e-3 | 1.6e-3 |
| Batch Size | 131,072 | 131,072 | 131,072 |
| Weight Decay | 0.01 | 0.01 | 0.01 |
| Max Steps | 8K | 128K | 128K |
| Learning rate Decay | Linear | Linear | Linear |
| LAMB $\epsilon$ | 1e-6 | 1e-6 | 1e-6 |
| LAMB $\beta_1$ | 0.9 | 0.9 | 0.9 |
| LAMB $\beta_2$ | 0.999 | 0.999 | 0.999 |
| Gradient Clipping | 0.0 | 0.0 | 0.0 |
| Sequence length ($T$) | 128 | 128 | 128 |

Table 8: The hyper-parameters for pretraining FOLNet in different settings.

## C  ADDITIONAL EXPERIMENTS

**Zero-shot performance on GLUE**  We evaluate the zero-shot performance of our FOLNet model on GLUE benchmark. Specifically, we perform zeros-shot predictions by using the same method and prompt templates from Gao et al. (2021). We only consider the FOLNet$_\text{Large}$ model and compare it to RoBERTa$_\text{Large}$, which are similar in model-size, pretraining dataset and pretraining losses. In

| Hyperparams | GLUE-small/big | SQuAD 2.0 | FOLIO |
|---|---|---|---|
| Max epochs | {5, 10, 20} / {2, 3, 5} | {2, 3} | {60, 80} |
| Peak lr for Base (16G): | {6e-5, 8e-5 1e-4, 2e-4} | { 8e-5, 9e-5, 1e-4 } | { 6e-5, 7e-5, 8e-5} |
| Peak lr for Base/Large (160G): | {1e-5, 2e-5, 3e-5, 4e-5} | {1e-5, 2e-5, 3e-5, 4e-5} | { 1e-5, 2e-5, 3e-5} |
| Batch size | {16, 32} / {32} | {16, 32} | {16, 32} |
| Learning rate decay | Linear | Linear | Linear |
| Warmup ratio | {6%, 25%} / {6%} | 6% | {6%, 25%} |
| Sequence length | 128 | 384 | 512 |
| Adam $\epsilon$ | 1e-6 | 1e-6 | 1e-6 |
| Adam $\beta_1$ | 0.9 | 0.9 | 0.9 |
| Adam $\beta_2$ | 0.999 | 0.999 | 0.999 |
| Gradient clipping | 0.0 | 0.0 | 0.0 |
| Dropout rate | 0.1 | 0.1 | 0.1 |
| Weight decay | 0.01 | 0.01 | 0.01 |

Table 9: The hyperparameters for finetuning on GLUE, SQuAD 2.0, and FOLIO tasks. GLUE-small refers to CoLA, STS-B, MRPC and RTE. GLUE-big stands for MNLI, QQP, QNLI and SST-2.

| Model | Method | Data | MNLI-m/mm Acc | QQP F1 | QNLI Acc | SST-2 Acc | CoLA mcc | STS-B Pear. | MRPC F1 | RTE Acc | Avg |
|---|---|---|---|---|---|---|---|---|---|---|---|
| Majority guess | - | - | 32.7/33.0 | 0.0 | 49.5 | 50.9 | 0.0 | - | 81.2 | 52.7 | 33.3 |
| RoBERTa$_{Large}$ | MLM | 160G | **50.8**/51.7 | 49.7 | 50.8 | **83.6** | 2.0 | -3.2 | 61.9 | 51.3 | 44.3 |
| FOLNet$_{Large}$ | MLM | 160G | **50.8**/52.6 | 55.4 | 59.7 | 79.5 | 6.4 | 23.6 | 77.2 | 58.2 | 51.5 |

Table 10: The zero-shot performance on the GLUE development set. "Acc" stands for accuracy, "mcc" means Matthews's correlation coefficient, and "Pear." is short for Pearson's correlation coefficient.

addition, the zero-shot prediction methods are also similar: they both predict the label words using their own MLM heads. The results are summarized in Table 10, which demonstrates that FOLNet$_{Large}$ outperforms RoBERTa$_{Large}$ on 7 out of 8 tasks with average gain of 7.2 points.

**Performance on CLUTRR** To further examine the reasoning capabilities, we evaluate our FOL-Net models on the CLUTRR (**C**ompositional **L**anguage **U**nderstanding and **T**ext-based **R**elational **R**easoning) dataset (Sinha et al., 2019). CLUTRR is a semi-synthetic diagnostic benchmark designed to evaluate the systematic generalization ability of a model. Specifically, for a given natural language story, a model is asked to infer the (implicit) relationship between two family members. To solve the problem, it has to extract relationships between entities and master the logical rules governing these relationships. CLUTRR examines the systematic generalization by testing a model on stories that contain unseen combinations of logical rules as well as stories that require more reasoning steps. Following the same setting as Sinha et al. (2019), we first finetune our FOLNet models with clauses of length $k = 2, 3$ and $k = 2, 3, 4$, respectively, and then we evaluate them on clauses of length $k = 2, \ldots, 10$. Specifically, for each input instance, we concatenate the natural language story with the text query (separated by a `[SEP]` token), and cast the problem as a sequence-pair classification. In addition, we add a `[CLS]` token at the beginning and append a `[SEP]` token in the end. However, unlike Sinha et al. (2019), for simplicity, we do not replace entities with special task-specific embeddings but treat them as regular English words. And we leave such entity representation techniques as a future work, which may further improve our performance. We finetune FOLNet$_{Base}$ for 100 epochs using an Adam optimizer with a learning rate of $5 \times 10^{-5}$, a batch-size of 16 and a warmup ratio of 6%. In our experiment, we mainly compare to transformer-based baselines that also use the *natural language inputs*. Specifically, we compare to the results of BERT$_{Base}$ and BERT-LSTM from Sinha et al. (2019), which share the same model size and pretraining corpus. There is also a rich set of approaches that work directly on the *symbolic inputs* from CLUTRR, such as the Graph Attention Network (GAT) (Veličković et al., 2018). Since these methods directly use the structured logical graph underlying the story as their input, they circumvent the difficulty of parsing natural language stories and generally perform much better than the text-based counterparts. We include GAT as a reference to examine whether our FOLNet model could narrow such a performance gap with the help of our logical inductive bias. The full results are reported in Figure 5. First, our FOLNet$_{Base}$ performs

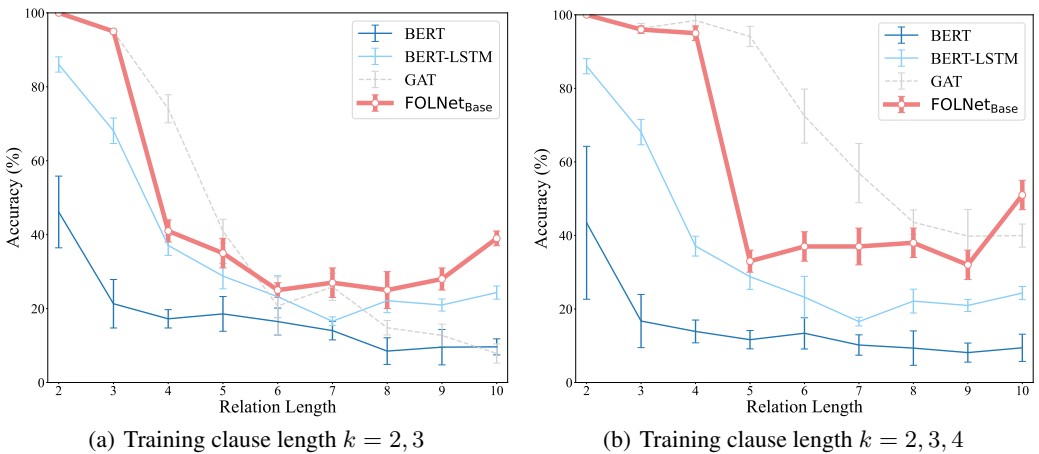

(a) Training clause length $k = 2, 3$               (b) Training clause length $k = 2, 3, 4$

Figure 5: Systematic generalization performance on CLUTRR benchmark.

significantly better than both text-based methods (i.e., BERT$_{\text{Base}}$ and BERT-LSTM) by a large margin. In particular, when the testing $k$ is out of the training set, we improve BERT$_{\text{Base}}$ by as much as $8\% \sim 41\%$ (in absolute improvement). Meanwhile, this advantage further improves to $54\% \sim 81\%$ when the testing $k$ has been seen during training, and FOLNet$_{\text{Base}}$ achieves competitive performance as GAT. Notably, FOLNet$_{\text{Base}}$ even achieves substantially better performance ($10\% \sim 20\%$) than GAT when the testing $k = 10$. The results confirms the benefit of encoding logical inductive bias.

## D   NEURAL MODUS PONENS

In this section, we show that generic Modus Ponens inference using clauses of the form (8) can be expressed as a matrix multiplication followed by an optional nonlinear activation function.

Suppose $P_1, \ldots, P_M$ are $M$ premise atoms and $Q$ is the head (i.e., conclusion) atom. We partition the set $\mathcal{M} = \{1, \ldots, M\}$ into three non-overlapping sets $\mathcal{M}_+$, $\mathcal{M}_0$ and $\mathcal{M}_-$, and define the following general clause (where we have dropped the subscript $n$ in (8) for simplicity of notation):

$$Q \leftarrow \left( \bigwedge_{m \in \mathcal{M}_+} P_m \right) \bigwedge \left( \bigwedge_{m \in \mathcal{M}_-} \neg P_m \right). \tag{11}$$

Note that the expression (11) can be used to represent a general clause that infers $Q$ from any subset of the premises in $\{P_1, \ldots, P_M, \neg P_1, \ldots, \neg P_M\}$, where the sets $\mathcal{M}_+$ and $\mathcal{M}_-$ contain the indexes of the selected original and negated premises, respectively, and the set $\mathcal{M}_0$ indexes the ignored premises. Therefore, each particular partition $\mathcal{M} = \mathcal{M}_+ \cup \mathcal{M}_0 \cup \mathcal{M}_-$ also corresponds to a unique clause. To apply Modus Ponens inference with the clause (11), we can proceed in two steps: (i) evaluate the body of the clause (i.e., the right-hand side of (11)) according to

$$P \equiv \left( \bigwedge_{m \in \mathcal{M}_+} P_m \right) \bigwedge \left( \bigwedge_{m \in \mathcal{M}_-} \neg P_m \right), \tag{12}$$

and (ii) apply the generic Modus Ponens inference rule: $Q \Leftarrow \{Q \leftarrow P \text{ and } P\}$. In the following two subsections, we will derive the neural operations that implement these two steps, respectively. Specifically, the neural operations will be carried out in the logit space under the ProbLog setting.

### D.1   THE MATRIX MULTIPLICATION FOR THE COMPOSITIONS OF THE BODY ATOMS

To derive the operations for the logic expression (12), we define the probabilities of the atoms $P, P_1, \ldots, P_M$ being true in the following logistic form:

$$\Pr\{P = \mathsf{T}\} = \frac{1}{1 + e^{-z}}, \qquad \Pr\{P_m = \mathsf{T}\} = \frac{1}{1 + e^{-v_m}}, \quad m = 1, \ldots, M, \tag{13}$$

where $z$ and $v_m$ are the logits corresponding to the atoms $P$ and $P_m$, respectively. In addition, for each $m$, we further introduce a variable $W_m = +1, 0, -1$ to indicate whether $m$ is in $\mathcal{M}_+$, $\mathcal{M}_0$ and $\mathcal{M}_-$, respectively. Then, (12) can be characterized by the following conditional probability:

$$\Pr\{P = \mathsf{T}|\mathcal{W}, \mathcal{P}\} = \begin{cases} 1 & \bigwedge_{m=1}^{M} \left[ (W_m = 1 \wedge P_m = \mathsf{T}) \vee (W_m = -1 \wedge P_m = \mathsf{F}) \vee (W_m = 0) \right] \\ 0 & \text{otherwise} \end{cases}, \quad (14)$$

where $\mathcal{W} \triangleq \{W_1, \ldots, W_M\}$ and $\mathcal{P} \triangleq \{P_1, \ldots, P_M\}$. We further assume that the variables $P_1, \ldots, P_M$ are independent of each other and are also independent of $W_1, \ldots, W_M$. In addition, for a given set of $\mathcal{W} = \{W_1, \ldots, W_M\}$, we introduce the following random event of $P_1, \ldots, P_M$:

$$\mathcal{E} \triangleq \left\{ (P_1, \ldots, P_M) : \bigwedge_{m=1}^{M} \left[ (W_m = 1 \wedge P_m = \mathsf{T}) \vee (W_m = -1 \wedge P_m = \mathsf{F}) \vee (W_m = 0) \right] \right\}, \quad (15)$$

along with its indicator function $\mathbb{I}((P_1, \ldots, P_M) \in \mathcal{E})$. An indicator function $\mathbb{I}(\cdot)$ is one if the expression inside its parenthesis is true and zero otherwise. Then, we can derive $\Pr\{P = \mathsf{T}|\mathcal{W}\}$ as:

$$\Pr\{P = \mathsf{T}|\mathcal{W}\} = \sum_{\mathcal{P}} \Pr\{P = \mathsf{T}|\mathcal{W}, \mathcal{P}\} \Pr\{P_1, \ldots, P_M|\mathcal{W}\}$$

$$\overset{(a)}{=} \sum_{\mathcal{P}} \Pr\{P = \mathsf{T}|\mathcal{W}, \mathcal{P}\} \Pr\{P_1, \ldots, P_M\}$$

$$\overset{(b)}{=} \sum_{\mathcal{P}} \mathbb{I}((P_1, \ldots, P_M) \in \mathcal{E}) \Pr\{P_1, \ldots, P_M\}$$

$$\overset{(c)}{=} \sum_{\mathcal{P}} \mathbb{I}((P_1, \ldots, P_M) \in \mathcal{E}) \prod_{m=1}^{M} \Pr\{P_m\}$$

$$\overset{(d)}{=} \sum_{(P_1, \ldots, P_M) \in \mathcal{E}} \prod_{m \in \mathcal{M}_+} \Pr\{P_m\} \prod_{m \in \mathcal{M}_-} \Pr\{P_m\} \prod_{m \in \mathcal{M}_0} \Pr\{P_m\}$$

$$\overset{(e)}{=} \sum_{(P_1, \ldots, P_M) \in \mathcal{E}} \prod_{m \in \mathcal{M}_+} \Pr\{P_m = \mathsf{T}\} \prod_{m \in \mathcal{M}_-} \Pr\{P_m = \mathsf{F}\} \prod_{m \in \mathcal{M}_0} \Pr\{P_m\}$$

$$\overset{(f)}{=} \prod_{m \in \mathcal{M}_+} \Pr\{P_m = \mathsf{T}\} \prod_{m \in \mathcal{M}_-} \Pr\{P_m = \mathsf{F}\} \sum_{\{P_m : m \in \mathcal{M}_0\}} \prod_{m \in \mathcal{M}_0} \Pr\{P_m\}$$

$$\overset{(g)}{=} \prod_{m \in \mathcal{M}_+} \Pr\{P_m = \mathsf{T}\} \prod_{m \in \mathcal{M}_-} \Pr\{P_m = \mathsf{F}\}$$

$$\overset{(h)}{=} \prod_{m \in \mathcal{M}_+} \frac{1}{1 + e^{-v_m}} \prod_{m \in \mathcal{M}_-} \frac{1}{1 + e^{v_m}}$$

$$\overset{(i)}{=} \prod_{m=1}^{M} \frac{1}{(1 + e^{-v_m})^{I_m^+}} \prod_{m=1}^{M} \frac{1}{(1 + e^{v_m})^{I_m^-}}$$

$$= \prod_{m=1}^{M} \frac{1}{(1 + e^{-v_m})^{I_m^+}} \cdot \frac{1}{(1 + e^{v_m})^{I_m^-}}$$

$$\overset{(j)}{\leq} \frac{1}{1 + e^{-\sum_{m=1}^{M}(I_m^+ - I_m^-)v_m}} \qquad (16)$$

where step (a) uses the independence between $P_1, \ldots, P_M$ and $W_1, \ldots, W_M$, step (b) substitutes the definition of the conditional probability (14), step (c) uses the assumption that $P_1, \ldots, P_M$ are independent of each other, step (d) substitutes the decomposition $\mathcal{M} = \mathcal{M}_+ \cap \mathcal{M}_0 \cap \mathcal{M}_-$, step (e) is obtained by following the definition (15) for the event $\mathcal{E}$ (i.e., within event $\mathcal{E}$, we have $P_m = \mathsf{T}$ for $m \in \mathcal{M}_+$, $P_m = \mathsf{F}$ for $m \in \mathcal{M}_-$ and $P_m$ being arbitrary value in $\{\mathsf{T}, \mathsf{F}\}$), step (f) takes out the common factors of the summands and keeps the remaining summation over all possible values in $\{P_m : m \in \mathcal{M}_0\}$ (based on the definition (15)), step (g) uses the fact that the total probability of the event $\{P_m : m \in \mathcal{M}_0\}$ is one, step (h) substitutes the second logistic expression in (13), step

(i) introduces variables $I_m^+ \triangleq \mathbb{I}(m \in \mathcal{M}_+)$ and $I_m^- \triangleq \mathbb{I}(m \in \mathcal{M}_-)$, and the upper bound in step (j) is obtained by expanding the multiplications in the denominator and dropping all the cross-terms, all of which are nonnegative. Note that the upper bound (16) is tight when only one unique $I_m^+$ or $I_m^-$ (among all $I_1^+, \ldots, I_M^+$ and $I_1^-, \ldots, I_M^-$) is nonzero. The above result in (16) is conditioned on a particular realization of $W_1, \ldots, W_M$, which are discrete variables. Therefore, it is difficult to learn them directly in a differentiable manner. To address this issue, we further assume that $W_1, \ldots, W_M$ are random variables. Then, taking expectation over $W_1, \ldots, W_M$, we have:

$$\Pr\{P = \mathsf{T}\} \leq \mathsf{E}\left\{\frac{1}{1 + e^{-\sum_{m=1}^M (I_m^+ - I_m^-)v_m}}\right\}, \tag{17}$$

where the randomness inside the expectation comes from $I_m^+$ and $I_m^-$, which are further from $W_1, \ldots, W_M$. We now adopt a simple yet effective strategy to approximate the right-hand side in order to obtain a differentiable implementation. Specifically, we use the following approximation:

$$\mathsf{E}\left[\frac{1}{1 + e^X}\right] \approx \frac{1}{1 + e^{\mathsf{E}X}}, \tag{18}$$

which becomes more accurate when the distribution of $X$ gets more concentrated around a single peak (i.e., becoming determinisitic). Using the above approximation, we obtain

$$\Pr\{P = \mathsf{T}\} \approx \frac{1}{1 + e^{-\sum_{m=1}^M (\kappa_m^+ - \kappa_m^-)v_m}} \tag{19}$$

where $\kappa_m^+ \triangleq \mathsf{E}[I_m^+] = \Pr\{W_m = +1\}$ and $\kappa_m^- \triangleq \mathsf{E}[I_m^-] = \Pr\{W_m = -1\}$. Substituting the first expression in (13) into the left-hand side of (19), we conclude that the logit of $\Pr\{P = \mathsf{T}\}$ can be approximated via:

$$z = \sum_{m=1}^M (\kappa_m^+ - \kappa_m^-)v_m = \langle \boldsymbol{\kappa}, \mathbf{v} \rangle \tag{20}$$

where $\boldsymbol{\kappa}$ and $\mathbf{v}$ denote the vectors that collect $\kappa_m^+ - \kappa_m^-$ and $v_m$ as their $m$-th elements, respectively. The above expression implies that the logit $z$ for the atom $P$ can be computed by a simple inner product operation between the two vectors $\boldsymbol{\kappa}$ and $\mathbf{v}$. Further recall that different realizations of $W_1, \ldots, W_M$ correspond to different partitions of $\mathcal{M} = \mathcal{M}_+ \cup \mathcal{M}_0 \cup \mathcal{M}_-$, which further defines different logic expressions for $P$ in (12). Since $\boldsymbol{\kappa}$ is a vector that collects $\Pr\{W_m = +1\} - \Pr\{W_m = -1\}$ as its $m$-th element, it can also be viewed as a signature vector of the logic expression (12), which is the body of the clause (11). Therefore, $\boldsymbol{\kappa}$ is also the signature vector for the clause in (11) as it determines the logic compositions among the body atoms. When we have multiple logic expressions, represented by $\boldsymbol{\kappa}_1, \ldots, \boldsymbol{\kappa}_N$, that compose $N$ logic expressions from the atoms $P_1, \ldots, P_M$, then we can compute the logits of the output logic expressions by the following matrix multiplication:

$$\mathbf{z} = \mathbf{K}\mathbf{v}, \tag{21}$$

where $\mathbf{z}$ is a logit vector for the $N$ output atoms, and $\mathbf{K}$ is a matrix consists of $\boldsymbol{\kappa}_n$ as its $n$-th row.

## D.2 THE NONLINEAR ACTIVATION FOR THE IMPLICATION OPERATION

In this subsection, we proceed to derive the the neural operator for generic Modus Ponens inference, $Q \Leftarrow \{Q \leftarrow P \text{ and } P\}$, in the logit space. Likewise, we consider the ProbLog setting, where the atoms $P$ and $Q$ will be assigned with probabilities $\Pr\{P = \mathsf{T}\}$ and $\Pr\{Q = \mathsf{T}\}$, respectively, which characterize the chances of them being true. Let $C \triangleq (Q \leftarrow P)$ be the clause. Our objective is to infer the outcome probability $\Pr\{Q = \mathsf{T}\}$ from $\Pr\{P = \mathsf{T}\}$ based on the fact that $C = \mathsf{T}$.

To this end, we first derive the conditional probability $\Pr\{Q = \mathsf{T}|P, C = \mathsf{T}\}$. We will first need to establish the conditional probability $\Pr\{C = \mathsf{T}|P, Q\}$ based on the definition of the logical implication "$\leftarrow$" (Andrews, 2013) and then apply Bayes rule. Note that $C = (Q \leftarrow P)$ is defined as $C \triangleq (Q \vee \neg P)$, which can be expressed as the following conditional probability:

$$\Pr\{C = \mathsf{T}|P, Q\} = \begin{cases} 0 & \text{if } P = \mathsf{T} \text{ and } Q = \mathsf{F} \\ 1 & \text{otherwise} \end{cases}. \tag{22}$$

That is, the clause $C$ will be false only when the premise $P$ is true and the conclusion $Q$ is false. To proceed, we further assume that $\Pr\{Q = \mathsf{T}|P\} = \Pr\{Q = \mathsf{F}|P\} = 1/2$. The intuition of the assumption is that we do not have any prior knowledge about the outcome $Q$ when we are only given the input premise $P$ (without $C$). Then, by Bayes rule, we have

$$
\begin{aligned}
\Pr\{Q = \mathsf{T}|P, C = \mathsf{T}\} &= \frac{\Pr\{Q = T, C = \mathsf{T}|P\}}{\Pr\{C = \mathsf{T}|P\}} \\
&= \frac{\Pr\{Q = \mathsf{T}|P\}\Pr\{C = \mathsf{T}|P, Q = \mathsf{T}\}}{\sum_{Q \in \{\mathsf{F}, \mathsf{T}\}} \Pr\{Q|P\}\Pr\{C = \mathsf{T}|P, Q\}} \\
&\overset{(a)}{=} \frac{\Pr\{C = \mathsf{T}|P, Q = \mathsf{T}\}}{\Pr\{C = \mathsf{T}|P, Q = \mathsf{T}\} + \Pr\{C = \mathsf{T}|P, Q = \mathsf{F}\}} \\
&\overset{(b)}{=} \begin{cases} 1 & \text{if } P = \mathsf{T} \\ 0.5 & \text{if } P = \mathsf{F} \end{cases},
\end{aligned}
\tag{23}
$$

where steps (a) and (b) substitute $\Pr\{Q = \mathsf{T}|P\} = \Pr\{Q = \mathsf{F}|P\} = 1/2$ and (22), respectively.

Next, we derive $\Pr\{Q = \mathsf{T}|C = \mathsf{T}\}$ with the assumption that the premise $P$ is independent of the clause $C$ that is used in the current Modus Ponens inference step. We have

$$
\begin{aligned}
\Pr\{Q = \mathsf{T}|C = \mathsf{T}\} &= \sum_{P \in \{\mathsf{T}, \mathsf{F}\}} \Pr\{Q = \mathsf{T}|P, C = \mathsf{T}\}\Pr\{P|C = \mathsf{T}\} \\
&\overset{(a)}{=} \sum_{P \in \{\mathsf{T}, \mathsf{F}\}} \Pr\{Q = \mathsf{T}|P, C = \mathsf{T}\}\Pr\{P\} \\
&\overset{(b)}{=} \Pr\{P = \mathsf{T}\} + \frac{1}{2}\Pr\{P = \mathsf{F}\} \\
&= \frac{1}{2} + \frac{1}{2}\Pr\{P = \mathsf{T}\},
\end{aligned}
\tag{24}
$$

where step (a) uses the assumption that $P$ is independent of $C$, and step (b) substitutes (23).

Finally, we derive the expression for the conditional probability (24) in the logit space. Specifically, let $q$ and $p$ be the logits for $Q$ and $P$, respectively, which parameterize their probabilities:

$$
\Pr\{P = \mathsf{T}\} = \frac{1}{1 + e^{-z}}, \qquad \Pr\{Q = \mathsf{T}|C = \mathsf{T}\} = \frac{1}{1 + e^{-u}}.
\tag{25}
$$

Substituting the first expression in (25) into (24) followed by some simple algebra, we obtain

$$
\Pr\{Q = \mathsf{T}|C = \mathsf{T}\} = \frac{1}{1 + e^{-\ln(1 + 2e^z)}}.
\tag{26}
$$

Comparing the right-hand side of (26) with that of (25), we obtain the logit for $P$ as

$$
u = \ln(1 + 2e^z).
\tag{27}
$$

vThe above expression implies that, to implement Modus Ponens in logit space, we only need to apply the above simple nonlinear activation function to the input premise logit $p$. To gain further insights into the above logit-space Modus Ponens inference, we carry out further analysis of the above nonlinear activation function. Note that $\ln(1 + e^x)$ is a non-negative convex function. By using Jensen's inequality and the non-negativity, we can prove that the above nonlinear activation function can be lower bounded as $\ln(1 + 2e^x) \geq \mathrm{ReLU}(x + \ln 2)$, where the right-hand side is indeed a good approximation of the left-hand side. Therefore, we can implement the generic Modus Ponens by applying the (shifted) ReLU function in the logit space with much lower computation complexity.

### D.3 Putting everything together

Given $N$ different clauses of the form (11) and $M$ input premises $P_1, \ldots, P_M$, the deduction of the $N$ outcome atoms $Q_1, \ldots, Q_N$ using Modus Ponens rule can be implemented (approximately) via

$$
\mathbf{z} = \mathbf{K}\mathbf{v}
\tag{28}
$$

$$\mathbf{u} = \ln(1 + 2e^{\mathbf{z}}), \tag{29}$$

where $\mathbf{K}$ is an $N \times M$ matrix with its $n$-th row being the signature vector of the $n$-th clause, $\mathbf{v}$ is an $M$-dimensional vector consisting of the logits for the input premises $P_1, \ldots, P_M$, $\mathbf{u}$ is an $N$-dimensional vector that contains the logits of the output atoms $Q_1, \ldots, Q_N$. Note that the Modus Ponens inference using $N$ clauses can be implemented in parallel by first multiply matrix $\mathbf{K}$ to the left of the logit vector $\mathbf{v}$ and then pass through a special element-wise nonlinear activation function (which can be approximated by a shifted ReLU function). For this reason, we call $\mathbf{K}$ the *kernel-of-clauses* (or *kernels* for short) in this paper. Furthermore, we note that, when the implication "$\leftarrow$" in (11) is replaced by the logical equivalence "$\equiv$", the original clause (11) becomes (12), so that the activation function in (29) can be dropped. Finally, it is straightforward to show that the neural Modus Ponens inference (28) for the four categories of rules in (7) can be expressed (equivalently) as:

$$\mathbf{u}(x) = \sum_a \mathbf{K}_{\mathtt{UU}}(x, a)\mathbf{v}(a) \tag{30}$$

$$\mathbf{u}(x) = \sum_{a,b} \mathbf{K}_{\mathtt{UB}}(x, a, b)\mathbf{v}(a, b) \tag{31}$$

$$\mathbf{u}(x, y) = \sum_a \mathbf{K}_{\mathtt{BU}}(x, y, a)\mathbf{v}(a) \tag{32}$$

$$\mathbf{u}(x, y) = \sum_{a,b} \mathbf{K}_{\mathtt{BB}}(x, y, a, b)\mathbf{v}(a, b), \tag{33}$$

where $\mathbf{K}_{\mathtt{UU}}(x, a)$, $\mathbf{K}_{\mathtt{UB}}(x, a, b)$, $\mathbf{K}_{\mathtt{BU}}(x, y, a)$ and $\mathbf{K}_{\mathtt{BB}}(x, y, a, b)$ are $D_1 \times D_1$, $D_1 \times D_2$, $D_2 \times D_1$ and $D_2 \times D_2$ matrices that correspond to $\mathcal{R}_{\mathtt{UU}}$, $\mathcal{R}_{\mathtt{UB}}$, $\mathcal{R}_{\mathtt{BU}}$ and $\mathcal{R}_{\mathtt{BB}}$, respectively.

## E    COMPOSITION OF EXISTING RELATIVE POSITIONAL ENCODING

We now show that we can compose the existing relative positional encoding (denoted as RPE$^\star$ in our paper) from our m-operator and p-operator. To begin with, we first write the expressions of existing RPE$^\star$ from Shaw et al. (2018) by using their original notation. Specifically, RPE$^\star$ computes the (multi-head) self-attention outputs according to the following expressions (with the notation of self-attention head $h$ being dropped for simplicity):

$$z_i = \sum_{j=1}^n \alpha_{ij}(x_j W^V + a_{ij}^V) = \sum_{j=1}^n \alpha_{ij} x_j W^V + \sum_{j=1}^n \alpha_{ij} a_{ij}^V \tag{34}$$

$$e_{ij} = \frac{x_i W^Q (x_j W^K + a_{ij}^K)^T}{\sqrt{d_z}} = \frac{x_i W^Q (x_j W^K)^T + x_i W^Q (a_{ij}^K)^T}{\sqrt{d_z}}, \tag{35}$$

where $z_i$ is the self-attention output at the $i$-th token, $e_{ij}$ is the unnormalized self-attention scores, $\alpha_{ij}$ is the self-attention probability (which is obtained by applying softmax to $e_{ij}$, normalized over $j$), $x_i$ is the vector of the $i$-th token at the attention input, $W^Q/W^K/W^V$ are the weight matrices for query/key/value in the self-attention operation, respectively, $n$ is the sequence length, and $d_z$ is the dimension of each head. Notably, $a_{ij}^V$ and $a_{ij}^K$ are the learnable relative positional embedding vectors, defined as

$$a_{ij}^K = w_{\mathrm{clip}(j-i,k)}^K$$
$$a_{ij}^V = w_{\mathrm{clip}(j-i,k)}^V$$
$$\mathrm{clip}(x, k) = \max(-k, \min(k, x)).$$

That is, $a_{ij}^V$ and $a_{ij}^K$ are the embedding vectors corresponding to the (clipped) relative distance between the $i$-th and the $j$-th tokens, where $k$ is the clipping threshold. Therefore, the second terms in both (34)–(35) are the RPE$^\star$ bias terms that seep into the computations of self-attention mechanism. We now proceed to show that these two terms can be viewed as the *degenerated* form of our m-operator and p-operator in Table 2. For convenience, we first rewrite the expressions for these two operators:

$$\mathsf{m}: \quad \mathbf{u}_{hs}(x) = \sum_a \mathbf{K}_h(x, a)\mathbf{v}_s(x, a) \tag{36}$$

$$\mathsf{p}: \quad \mathbf{u}_h(x,y) = \sum_w \mathbf{K}_{hw}(x)\mathbf{v}_w(x,y). \tag{37}$$

We first show that the second terms in (34) can be composed from the m-operator (36). To see this, note that tokens $x$ and $a$ in (36) can be identified as $i$ and $j$ in (34), respectively. In addition, the kernel $\mathbf{K}_h(x,a)$ and the premise $\mathbf{v}_s(x,a)$ can be identified as the self-attention probability $\alpha_{ij}$ and the relative positional embedding vector $a_{ij}^V$, respectively. Therefore, the m-operator shares the same form as the second term in (34), except our "head" index $h$. Likewise, we can show that the second term in (35) can be composed from the p-operator. Observe that the second term in (35) is an inner product between vector $x_i W^Q$ and vector $a_{ij}^K$. Let $\mathbf{k}_h(x)$ be a vector that collects $\mathbf{K}_{hw}(x)$ as its $w$-th element and let $\mathbf{v}(x,y)$ be a vector that collects $\mathbf{v}_w(x,y)$ as its $w$-th element. Then, the p-operator can also be viewed as an inner product between the vectors $\mathbf{k}_h(x)$ and $\mathbf{v}(x,y)$. By identifying $\mathbf{k}_h(x)$ and $\mathbf{v}(x,y)$ as the vectors $x_i W^Q$ and $a_{ij}^K$, respectively, we conclude that the second term in (35) can be composed from the p-operator. Besides these similarities, our m-operator and p-operator are more general and powerful than the original RPE$^\star$ in the following aspects. First, recall from Section 3.3 that both the kernels $\mathbf{K}$ and $\mathbf{v}$ are parametrized by the FOLNet (by linearly projecting the intermediate representations $\{\mathbf{u}_l(x), \mathbf{u}_l(x,y)\}$ followed by possible activation functions). Therefore, our $\mathbf{v}_s(x,a)$ and $\mathbf{v}_w(x,y)$ are *instance-dependent* and are different across input instances. In contrast, $a_{ij}^V$ and $a_{ij}^K$ in (34)–(35) are static embedding vectors that are the same for all input instances. Furthermore, our m-operator and p-operator are adaptive in a layerwise manner; that is, each layer will compute their own m-operator and p-operator adaptively based on their own intermediate representations $\{\mathbf{u}_l(x), \mathbf{u}_l(x,y)\}$, whereas in RPE$^\star$ the vectors $a_{ij}^V$ and $a_{ij}^K$ are generally identical across layers. Therefore, the existing operations in RPE$^\star$ can be viewed as the *degenerated* special cases that are composable from our m-operator and p-operator.

