# OpenReview forum: "Learning Language Representations with Logical Inductive Bias"
_ICLR.cc/2023/Conference — ICLR 2023 poster_

### Official Review · Reviewer_dLtS · 2022-10-19

**Confidence:** 2
**Clarity, Quality, Novelty And Reproducibility:** It is novel and of high quality.
**Correctness:** 3
**Technical Novelty And Significance:** 3
**Empirical Novelty And Significance:** 3
**Recommendation:** 6

**Strength And Weaknesses:**

Strength:

1. This paper develops a novel neural architecture named FOLNet (First-Order Logic Network), to encode this new logic inductive bias.

2. This paper finds that the self-attention module in transformers can be composed by two of our neural logic operators, which probably explains their strong reasoning performance.

3. The extensive experimental results support their  claims and validates the superiority.

Weaknesses：

1. It is not clear why FOLNet  could encode  logic inductive bias?

**Summary Of The Paper:**

This paper introduce a novel logical inductive bias which treats the language representation learning problem as logic programming. We then develop a fully-differentiable neural architecture (FOLNet) that forward-chains a set of neural logic operators, which effectively encodes this inductive bias. The proposed FOLNet architecture has the same input-output interface as the transformer models and can be pretrained over large-scale text data. Experimental results demonstrate that the FOLNet architecture significantly outperforms different variants of transformer models under the same pretraining losses. The results further show that the inherent dual-branch architecture has many advantages over the single-branch version in transformers.

**Summary Of The Review:**

Interesting and useful method.

---

> ### Author Response · Authors · 2022-11-19
> **Response to Reviewer dLtS**
>
> We thank the reviewer for the constructive feedback. We now address the question raised by the reviewer below. In addition, we have also incorporated it in our updated paper that we just uploaded.
>
> **Q1: Clarify how FOLNet encodes logical inductive bias.**
>
> Thanks for the suggestion. We will briefly clarify this point below. In addition, we have also slightly revised Section 3 of our papers to incorporate these clarifications. In a nutshell, our proposed FOLNet model encodes the logical inductive bias by performing a collection of Modus Ponens (MP) inference in the logit space, as we will explain below.
>
> First, we use a real-valued vector $\mathbf{u}(x_1,\ldots,x_r)$ to collect the logits of atoms, where the $d$-th element $u_d(x_1,\ldots,x_r)$ is the logit for the $d$-th atom $p_d(x_1,\ldots,x_r)$ so that the probability of it being true is expressed as:
> $$\mathrm{Pr}[ p_d(x_1,\ldots,x_r) = \mathsf{T} ] = \frac{1}{1+e^{-u_d(x_1,\ldots,x_r)}}.$$ Therefore, a larger  logit value implies higher chance of the atom being true and a negative value means it is less likely. Besides, we can also get the logit of the negated atom $\neg p_d(x_1,\ldots,x_r)$ by simply flipping the sign of $u_d(x_1, \ldots, x_r)$.
>
> Second, we implement the Modus Ponens inference in logit space as a matrix multiplication between a kernel matrix and a premise matrix (followed by an optional nonlinear activation function). The premise matrix contains the logits of all the premise atoms that could possibly be used, and the kernel matrix models a set of clauses that infers their corresponding head atoms in-parallel from the body atoms (represented by the premise matrix).  Please see our updated Sections 3.2 for more detailed clarifications. To control the complexity of the resulting matrix multiplication, we follow certain principles (Sec. 3.2) and enumerate all permissible matrix multiplications between the kernels and premises, which leads to the set of neural logic operators in Table 2. Notably, each neural logic operator with a particular form of matrix multiplication corresponds to a specific low-complexity Modus Ponens inference, where the conjunction operations (in the body of the clause) are restricted to be over a certain designated dimension. Please refer to our updated Appendix A.1 for more details regarding how each neural logic operator in Table 2 implements its own particular restrictions to model a low-complexity Modus Ponens inference. In addition, we also provide some examples in Appendix A.1 to demonstrate how the operators facilitate different aspects of reasoning.
>
> Finally, we forward chain these neural logic operators into a differentiable network, which deduces the logits of the output atoms from that of the input atoms.

---

### Official Review · Reviewer_93RW · 2022-10-24

**Confidence:** 3
**Correctness:** 3
**Technical Novelty And Significance:** 3
**Empirical Novelty And Significance:** 3
**Recommendation:** 8

**Clarity, Quality, Novelty And Reproducibility:**

The paper is generally well written, aside from some typographical nitpicks which I will list here. The experiments and mathematical formalism are quite thorough, and the comparisons are to widely used benchmarks. While this is not the first work to apply ideas from inductive logic programming to deep learning, they make a solid contribution.


Various copy-editing nits from experiment section:

“Learns better language representations” should be “learn better language representations”

“Many different variants of widely encoder-only” should be “many different variants of widely used encoder-only”

“To demonstrates its advantage” should be “to demonstrate its advantage”

“We first pretaining” should be “we first pretrain”


Issues with experiment presentation:

ALBERT XXL+ value of 96.9 is not bold but FOLNet Large of 96.8 is bold in the same column

Advantage of dual branch, Benefits of larger D2, contributions of larger D2, often are referencing table 4 when you mean to reference table 3

In 4.3, “line #14 on table 9” is referring to the wrong table, since table 9 is in an appendix and has no line #14. These are supposed to refer to table 4 I believe

In table #9 for MNLI, both 50.8’s should be bolded

**Strength And Weaknesses:**

Method:

The problem of learning new transformer architectures to aid reasoning is of broad interest to the community.

The model is well motivated by comparing to things like relative position encoding, and trying to use logical formalisms to derive new architectural variants on transformers.

How much the new architectural variants actually maintain the spirit of the logical formalisms is a bit harder to parse,  for example in 3.1 we are replacing discrete booleans with arbitrary real numbers and stating that “it characterizes the extent to which the atom is true”, which usually implies some value in [0,1] or some kind of increasing transformation, which I do not think you have here.

Experiments:

The experiments are quite thorough, but are often confounded by the proposed method having more parameters than the baselines. Additionally, experiments would be better if they included your own reruns of BERT/Roberta experiments using the same settings to make it apples to apples in some cases, this is especially helpful for the situations where gains are small. This would also make it easier to make apples-to-apples comparisons in terms of # of parameters, but it would be a lot of work.

The ablation experiments in Table #3 are nice and quite appreciated. It would be good to have another column saying how many parameters each ablated model has.

Having some more concrete examples / toy examples of how these new architectures specifically aid reasoning on concrete examples would be really helpful.



**Summary Of The Paper:**

The paper proposes a new family of transformer-style architectures inspired by inductive logic programming, with the goal of improving natural language reasoning capabilities. They connect this new family of architectures to existing commonly used modifications. They show good results on widely used benchmarks, modulo some concerns I raise later.

**Summary Of The Review:**

This is a well-motivated and well-executed paper addressing a problem of wide interest to the community. It could benefit from more experiments giving intuition of how the architectures work in practice, and some very careful apples-to-apples comparisons, but the ablations already go a good way in that direction. This paper merits acceptance.

---

> ### Author Response · Authors · 2022-11-19
> **Response to  Reviewer 93RW**
>
> We thank the reviewer for the constructive feedback. We now address the questions raised by the reviewer below. In addition, we have also incorporated them in our updated paper that we just uploaded.
>
> **Q1: How does the new architecture maintain the spirit of the logical formalisms? For example, the replacement of the discrete booleans with real numbers.**
>
> Thanks for the suggestion. We will briefly clarify this point below. In addition, we have also slightly revised Section 3 of our papers to incorporate these clarifications. In a nutshell, our proposed FOLNet model encodes the logical inductive bias by performing a collection of Modus Ponens (MP) inference in the logit space, as we will explain below.
>
> First, we use a real-valued vector $\mathbf{u}(x_1,\ldots,x_r)$ to collect the logits of atoms, where the $d$-th element $u_d(x_1,\ldots,x_r)$ is the logit for the $d$-th atom $p_d(x_1,\ldots,x_r)$ so that the probability of it being true is expressed as:
> $$\mathrm{Pr}[ p_d(x_1,\ldots,x_r) = \mathsf{T} ] = \frac{1}{1+e^{-u_d(x_1,\ldots,x_r)}}.$$ Therefore, a larger  logit value implies higher chance of the atom being true and a negative value means it is less likely. Besides, we can also get the logit of the negated atom $\neg p_d(x_1,\ldots,x_r)$ by simply flipping the sign of $u_d(x_1, \ldots, x_r)$.
>
> Second, we implement the Modus Ponens inference in logit space as a matrix multiplication between a kernel matrix and a premise matrix (followed by an optional nonlinear activation function). The premise matrix contains the logits of all the premise atoms that could possibly be used, and the kernel matrix models a set of clauses that infers their corresponding head atoms in-parallel from the body atoms (represented by the premise matrix).  Please see our updated Sections 3.2 for more detailed clarifications. To control the complexity of the resulting matrix multiplication, we follow certain principles (Sec. 3.2) and enumerate all permissible matrix multiplications between the kernels and premises, which leads to the set of neural logic operators in Table 2. Notably, each neural logic operator with a particular form of matrix multiplication corresponds to a specific low-complexity Modus Ponens inference, where the conjunction operations (in the body of the clause) are restricted to be over a certain designated dimension. Please refer to our updated Appendix A.1 for more details regarding how each neural logic operator in Table 2 implements its own particular restrictions to model a low-complexity Modus Ponens inference. In addition, we also provide some examples in Appendix A.1 to demonstrate how the operators facilitate different aspects of reasoning.
>
> Finally, we forward chain these neural logic operators into a differentiable network, which deduces the logits of the output atoms from that of the input atoms.
>
> **Q2: Include your own reruns of BERT/RoBERTa experiments using the same settings.**
>
> Based on your suggestion, we have pretrained a BERT model by using the same settings of FOLNet (e.g., large-batch pretraining with LAMB optimizer) so that we can carry out an apple-to-apple comparison. The results are reported in line #2 of Table 3 in the revised paper. Our reproduced BERT results are comparable to the ones reported in literature. The slightly degraded average performance might be caused by large-batch pretraining, LAMB optimizers, or the shorter pretraining sequence length (i.e., 128 throughout our work compared to 512 in literature).
>
> **Q3: Add another column in Table 3 to include the number of parameters of each ablated model.**
>
> Based on your suggestion, in the updated paper, we have added a new column into Table 3 to include the number of parameters for each ablated model. Observe that our proposed RPE slightly saves model parameters compared to APE and existing RPE (i.e., RPE*). In addition, adding more logic operators also slightly increases the number of model parameters, which is not surprising. However, in return, they will greatly improve the overall performance.
>
> **Q4: Having some more concrete examples / toy examples of how these new architectures specifically aid reasoning on concrete examples would be really helpful.**
>
> Thanks for the suggestion. In our updated paper, we have added some concrete (toy) examples in (the 2nd paragraph of) Appendix A.1 to demonstrate how each of the proposed logic operator in Table 2 help reasoning in these cases.
>
> **Q5: Typos and minor edits.**
>
> Thanks for pointing them out. We have fixed all these typos in our revised paper and have also thoroughly proof-read the entire paper again.

---

### Official Review · Reviewer_nDcE · 2022-10-24

**Confidence:** 4
**Correctness:** 4
**Technical Novelty And Significance:** 4
**Empirical Novelty And Significance:** 4
**Recommendation:** 8

**Clarity, Quality, Novelty And Reproducibility:**

The paper is clear and easy to read.
Experimental details are clear and should be reproducible.
Some methodology clarifications are needed (see section above).


**Strength And Weaknesses:**

**strengths**

This is a strong paper that introduces a novel architecture with inductive biases from first-order logic. Experimental results are very promising and the fact that logical tasks remain a challenge for traditional Transformers can make this work impactful.
The paper is well written and surprisingly easy to read. However the methodology could benefit from some clarifications listed below:

**weaknesses**

1. Methodology : After pre-training on large datasets, were the models fine-tuned on individual tasks (GLUE, SQuAD, FOLIO) ? If so, for how long? it should be mentioned somewhere.

2. Methodology : It is not clear how the final representations output from the model u_L(x) and u_L(x, y) are used in the computation of any loss. How does one go from these two representations to say a softmax layer to predict the masked token in MLM? Similarly for the other losses NSP and SOP.

*Question*: Related to the point above, is it safe to assume that FOLNets are encoder type networks and cannot be used in an decoder-only or encoder-decoder fashion? If this assumption is wrong, then the paper should also describe how u_L(x) and u_L(x, y) are used to predict next words.

3. Literature : Some influential previous work should be further discussed. Neural Theorem Provers (NeurIPS’17), and in particular Greedy Neural Theorem Provers (AAAI’20) applied logical modules on text before. More recently, Edge Transformers (NeurIPS’21) proposed a triangular attention mechanism between pairs of tokens in an effort to better represent logical behaviors between tokens. Comparing FOLNets to these works would help contextualize the work better.

4. Experiments : Right now the paper shows improvements on “classical” language tasks. It would be a nice bonus to also show that the FOL inductive bias is indeed useful for more reasoning heavy tasks such as GSM8k, ProofWriter, CLUTRR, etc…


**Summary Of The Paper:**

This work introduces a new neural architecture (FOLNet) that incorporates a first-order logical inductive bias. Taking inspiration from the forward-chaining algorithm, the architecture is recursively applying learnable Kernel operations on unary and binary representations of tokens.

Input and outputs are traditional token ids, making FOLNet an easy plug-and-play replacement from traditional Transformers. In addition, traditional Transformers can still be represented by a FOLNet making the proposed architecture more flexible and likely more powerful.

Experiments on GLUE, SQuAD2.0, and FOLIO show that this new architecture performs better than comparable baselines such as BERT, RoBERTa, ALBERT, and Megatron.


**Summary Of The Review:**

strong paper. more previous work should be discussed and some clarifications are needed.

---

> ### Author Response · Authors · 2022-11-19
> **Response to Reviewer nDcE (1/3)**
>
> We thank the reviewer for the constructive feedback. We now address the questions raised by the reviewer below. In addition, we have also incorporated them in our updated paper that we just uploaded.
>
> **Q1: Were the pretrained models finetuned on individual tasks? If so, for how long.**
>
> Yes. In all the downstream tasks, we finetune the pretrained models on individual tasks in order to fairly compare to other baselines. Our pretrained models could also be finetuned in a multi-task manner if needed. In our revised paper, we add a sentence (before the last sentence of Sec. 4.1) to clarify this point.
>
> Different tasks are finetuned for different numbers of epochs, depending on the task size (e.g., 2-3 epochs for SQuAD and 2-20 epochs for different GLUE tasks). We have reported all the detailed finetuning hyper-parameters in Table 8 of Appendix B.2, where we follow the standard procedure of hyper-parameter search as in RoBERTa and ALBERT. For the time consumption of finetuning, it is much less than the pretraining even with only 4-8 GPUs. For example, the GLUE tasks take 0.1-3 hours for FOLNet-base and 0.2-7 hours for FOLNet-large. The SQuAD-2.0 task takes 4.3 hours for FOLNet-base and takes 8 hours for FOLNet-large. The FOLIO task takes 1.2 hours for FOLNet-base and takes 3 hours for FOLNet-large.
>
> **Q2: How are the final representations output from $\mathbf{u}_L(x)$ and $\mathbf{u}_L(x,y)$ used in computing losses (e.g., MLM, NSP, SOP)?**
>
> As we pointed out in the paper, FOLNet models will have a similar input-output interface as the existing transformer models, so that we can seamless adopt existing pretraining losses (e.g., MLM, NSP and SOP) by computing them from $\{\mathbf{u}_L(x)\}$. Recall that $\mathbf{u}_L(x_t)$ is the vector that represents the derived (advanced) properties for the object (token) $x_t$, where $t=1, \ldots ,T$. For a masked token $x_t$, $\mathbf{u}_L(x_t)$ contains its properties that could be deduced from other tokens via their input properties and relations. Therefore, these deduced properties in $\mathbf{u}_L(x_t)$ can be used to predict the original masked token. For example, we can apply a linear classifier followed by a softmax operator to compute a probability distribution over the vocabulary and the MLM loss. Likewise, to compute NSP and SOP losses, we add a special “[CLS]” token at the beginning of the input sequence, so that the $\mathbf{u}_L(x)$ that corresponds to the “[CLS]” token will be fed into a binary classifier for computing the NSP or SOP loss. The usage of $\{\mathbf{u}_L(x_t)\}$ in downstream tasks (such as sequence classification, multiple-choice and sequence labeling) are also similar.
>
> On the other hand, we have not used the output binary predicates $\{\mathbf{u}_L(x_t,y_\tau)\}$ for computing any losses. The main reason is that we would like to adopt the existing off-the-shelf pretraining losses for an apple-to-apple comparison regarding the proposed model architecture. Since most of these losses are developed for the transformer architecture, which is a single-branch model with only unary predicates on its main pathway (Figure 2), it is not surprising that these losses are mainly computed from the unary properties $\{\mathbf{u}_L(x_t)\}$. Nevertheless, we believe that our newly introduced binary predicate branch could open a new avenue for developing additional pretraining losses using $\mathbf{u}_L(x_t,y_\tau)$ (i.e., the token relations). For example, we may randomly swap two tokens $x_t$ and $x_\tau$ and use $\mathbf{u}_L(x_t, x_\tau)$ to predict whether they have been swapped or not. We will leave the development of more effective pretraining losses for FOLNet as a future work.
>
> In the revised paper, we have added a short Appendix A.4 to clarify the above point.

---

> > ### Author Response · Authors · 2022-11-19
> > **Response to Reviewer nDcE (2/3)**
> >
> > **Q3: Are FOLNet encoder type networks and cannot be used in an decoder-only or encoder-decoder fashion? If this is not true, describe how $\mathbf{u}_L(x)$ and $\mathbf{u}_L(x,y)$ are used to predict next words.**
> >
> > Although we have mainly focused on the encoder-only version of the FOLNet model, it can be extended to the decoder-only and the encoder-decoder counterparts in a relatively straightforward manner. In our updated paper, we have added a new Appendix A.5 to discuss how to do such extension with more details. We now briefly explain the key ideas below.
> >
> > To develop the **decoder-only** version of FOLNet, we need to let the model auto-regressively generate the output tokens. In the context of FOLNet, this requires the unary and binary atoms $\{\mathbf{u}(x_t), \mathbf{u}(x_t, x_{\tau})\}$ to be inferred only from the atoms in the past. In addition, we further restrict the binary atoms to have a causal structure, i.e., $\mathbf{u}(x_t, x_\tau) = 0$ whenever $t < \tau$. We enforce such auto-regressive property by multiplying a proper 0-1 mask to the binary kernels and premises in the neural logic operators. In the new Appendix A.5 of our updated paper, we discuss how to adjust the model by applying these masks. Specifically, we are able to convert FOLNet into two typical variants of decoder-only models:  (i) Causal Langauge Model and (ii) Prefix Langauge Model. With such simple modifications, our decoder-only FOLNet model will have the same input-output interface as the decoder-only transformers; it can be pretrained to predict the next tokens using a linear classifier over the unary atoms $\mathbf{u}_L(x_t)$. After the pretraining, it can generate tokens in an auto-regressive manner.
> >
> > For the **encoder-decoder** variant, we use two separate stacks of FOLNet for the encoder and decoder, where each of them has the same overall architecture as in Figure 1 with its own set of atoms. In particular, the encoder will be identical to the encoder-only version that we have thoroughly discussed in the paper. Meanwhile, the decoder part will be similar to the decoder-only variant with a few additional modifications. First, the decoder needs to maintain a slightly different version of binary atoms. Specifically, besides the relations between the output tokens, the decoder also has to model the (unidirectional) relations from the input tokens to the output tokens. These relations are crucial in deducing the output tokens from the input ones, which plays a similar role as the cross-attention scores in transformers. Accordingly, we also adjust the masks to handle these relations separately. Second, the neural logic operators in Table 2 should also be slightly adjusted in order to incorporate the additional premise atoms from the encoder output. In our new Appendix A.5, we discuss the details of these changes. Notably, the encoder-decoder version of FOLNet retains the dual-branch architecture in its decoder module as well, which is in sharp contrast to the single-branch architecture of the transformer decoders.

---

> > > ### Author Response · Authors · 2022-11-19
> > > **Response to Reviewer nDcE (3/3)**
> > >
> > > **Q4: Incorporate and discuss more previous works: Neural Theorem Provers (NeurIPS’17), Greedy Neural Theorem Provers (AAAI’20), Edge Transformers (NeurIPS’21).**
> > >
> > > Thanks for mentioning these related works. We have added them to our revised paper and further discuss them in the revised Sec. 5. Meanwhile, we will also discuss them below.
> > >
> > > Both Neural Theorem Prover (NTP) and Greedy Neural Theorem Prover (GNTP) focus on the problem of proving queries to knowledge base (KB); they seek to prove a structured query by using the facts and rules from a KB. NTP adopts a backward chaining strategy to recursively construct a differentiable neural network using three neural modules (unification, OR and AND), where the discrete unification between atoms is approximated by a differentiable operator that computes the similarities between their embeddings. GNTP further reduces the complexity of NTP by pruning the proof paths using techniques such as nearest neighbor search. In contrast, we develop our FOLNet architecture based on the forward-chaining strategy, with the objective of learning better language representations from texts. Although GNTP is also capable of jointly reasoning over KB and texts, it is limited to the textual mentions that link two entities, which are treated as extra relation types and are encoded into relation embeddings to be processed in the same way as other relations. Therefore, it still falls under the same umbrella of KB reasoning.
> > >
> > > Similar to our work, the Edge Transformer also recognizes the importance of learning dependencies between relations, and develops a triangular attention mechanism that updates relations from other relations. In our context, it can be viewed as a single-branch architecture that focuses on the reasoning type of $\mathbf{u}(x,y) \Leftarrow \mathcal{R}_{\mathtt{BB}}, \mathbf{v}(a, b)$. In contrast, our FOLNet is a dual-branch architecture that performs reasoning over both unary properties and binary relations via a rich set of logic operators. Besides, the triangular attention mechanism generally has high storage and computation complexity (e.g., cubic in sequence length), whereas FOLNet has the same (quadratic) complexity as standard transformers.
> > >
> > > **Q5: It would be a nice bonus to also show the FOL inductive bias is indeed useful for more reasoning heavy tasks.**
> > >
> > > Thanks for the suggestion. In our updated paper, we have included a new experiment on the CLUTRR dataset in Appendix C. The results also show the advantage of FOLNet compared to the classical transformer-based pretrained models.
> > >
> > > In our original submission, besides the general language understanding tasks (GLUE and SQuAD), we have also used FOLIO as a task to evaluate the (first-order) reasoning capability of FOLNet. FOLIO is a challenging natural language reasoning dataset written by expert annotators based on real-world wikipedia pages (see Table 7 in Appendix B.1 for an example). It consists of first-order logical reasoning problems that have abundant natural language variations, a rich vocabulary, as well as diverse logic patterns. Another reason for choosing FOLIO as a reasoning benchmark is that it includes the knowledge necessary for performing reasoning in its input premises, so that we can focus completely on examining the reasoning capability. The results (in Table 4) show that FOLNet demonstrates much stronger first-order logic reasoning capability (e.g., $+3.9$ over RoBERTa-Large).
> > >
> > > We have not used GSM8k because it requires extra mathematical knowledge (that is generally missing in the problem context) in order to perform reasoning. The current state-of-the-art methods use a very large amount of model parameters (e.g., 175B in GPT-3 and 540B in PaLM & Minerva) to implicitly store such mathematical knowledge. To be comparable, we will have to pretrain a much larger model than our current FOLNet-Large (0.4B) in order to encode such knowledge (implicitly) into the model parameters. Besides, the GSM8k and ProofWriter tasks require a decoder-only or encoder-decoder model to generate the answers. Although our model can be easily extended to the decoder-only / encoder-decoder version (as we have discussed earlier), our current FOLNet models are only pretrained in the encoder-only fashion and are hence not directly applicable to such scenarios. Therefore, we will leave such an extension to future work.

---

> > > > ### Comment · Reviewer_nDcE · 2022-11-21
> > > > **response to authors**
> > > >
> > > > Thank you for all the additional information you provided. This makes the paper complete and is an accept for me.

---

### Decision · Program_Chairs · 2023-01-20

**Decision:**

Accept: poster

**Justification For Why Not Higher Score:**

The proposed approach isn't as groundbreaking as the scores as. Although it's inspired by first-order logic, it's still a blackbox machinery and doesn't perform actual reasoning.

**Justification For Why Not Lower Score:**

The idea of mimicking first-order logic reasoning in the neural architecture design is refreshing, so I would recommend acceptance.

**Metareview: Summary, Strengths And Weaknesses:**

This paper proposes a new neural architecture inspired by first-order logic and the authors have conducted extensive experiments to show the effectiveness of their approach.

I do share the concern with Reviewer dLtS: While the model architecture is inspired by first-order logic, there is no evidence that the model is performing ACTUAL first-order logic reasoning. The proposed model is still a black box and it's not completely clear what kind of inductive bias that the architecture brings in.

**Note From Pc:**

if the above contains the word "oral" or "spotlight" please see: "oral" presentation means -> notable-top-5% and "spotlight" means -> notable-top-25%. As stated in our emails, we are disassociating presentation type from AC recommendations

**Summary Of Ac-Reviewer Meeting:**

We did not have a meeting as all reviewers unanimously recommend this paper.